# Breaking Barriers in Eye Treatment: Polymeric Nano-Based Drug-Delivery System for Anterior Segment Diseases and Glaucoma

**DOI:** 10.3390/polym15061373

**Published:** 2023-03-09

**Authors:** Kevin Y. Wu, Said Ashkar, Shrieda Jain, Michael Marchand, Simon D. Tran

**Affiliations:** 1Department of Surgery, Division of Ophthalmology, University of Sherbrooke, Sherbrooke, QC J1G 2E8, Canada; yang.wu@usherbrooke.ca (K.Y.W.);; 2Faculty of Medicine, University of Ottawa, Ottawa, ON K1H 8M5, Canada; 3Department of Experimental Surgery, McGill University, Montreal, QC H3G 1A4, Canada; 4Faculty of Dental Medicine and Oral Health Sciences, McGill University, Montreal, QC H3A 1G1, Canada

**Keywords:** polymeric nanocarriers, biodegradable polymers, polymeric biomaterials, anterior segment diseases, glaucoma, ocular diseases, ocular drug-delivery

## Abstract

The eye has anatomical structures that function as robust static and dynamic barriers, limiting the penetration, residence time, and bioavailability of medications administered topically. The development of polymeric nano-based drug-delivery systems (DDS) could be the solution to these challenges: it can pass through ocular barriers, offering higher bioavailability of administered drugs to targeted tissues that are otherwise inaccessible; it can stay in ocular tissues for longer periods of time, requiring fewer drug administrations; and it can be made up of polymers that are biodegradable and nano-sized, minimizing the undesirable effects of the administered molecules. Therefore, therapeutic innovations in polymeric nano-based DDS have been widely explored for ophthalmic drug-delivery applications. In this review, we will give a comprehensive overview of polymeric nano-based drug-delivery systems (DDS) used in the treatment of ocular diseases. We will then examine the current therapeutic challenges of various ocular diseases and analyze how different types of biopolymers can potentially enhance our therapeutic options. A literature review of the preclinical and clinical studies published between 2017 and 2022 was conducted. Thanks to the advances in polymer science, the ocular DDS has rapidly evolved, showing great promise to help clinicians better manage patients.

## 1. Introduction

The eye is a complex and delicate organ that is protected by robust anatomical barriers that limit the penetration, bioavailability, and residence time of drugs administered topically. To overcome these challenges, the development of polymeric nano-based drug-delivery systems (DDS) has emerged as a promising solution. These systems can pass through ocular barriers and provide higher bioavailability of administered drugs to targeted tissues, leading to improved therapeutic outcomes. Furthermore, the use of biodegradable polymers minimizes the adverse effects of administered drugs that are not able to decompose naturally, such as the risk of infection, tissue damage, or reaction to toxic byproducts.

In recent years, there has been a growing body of research focused on exploring the therapeutic potential of polymeric nano-based DDS for ocular diseases. This review article provides a comprehensive overview of the current state of the field, including the material science aspects of the biopolymers used in DDS. It also highlights the key findings of preclinical and clinical studies conducted between 2017 and 2022. Our review highlights the therapeutic innovations and advances in polymer science that have contributed to the rapid evolution of ocular DDS, and how these systems are well positioned to help clinicians better manage patients with anterior segment diseases and glaucoma particularly. To provide a well-rounded review, this article will also cover non-polymeric nano-based DDS for ocular diseases, comparing and contrasting their features with those of polymeric nano-based DDS.

By providing a comprehensive overview of the latest findings in the field, this article provides insights into the current state of the art and highlights the future direction of research in this area.

## 2. Overview of Polymeric Nano-Based Drug-Delivery System (DDS)

### 2.1. Exploring Nanopolymers to Address Limitations of Traditional Formulations

Repetitive drug administrations with high dosages are often required for conventional formulations due to their extremely low bioavailability and fast clearance. This results in reduced patient compliance and can increase systemic side effects. Nano-based drug-delivery systems have the potential to reduce the frequency of administration for eye drops, as well as combine the administration of multiple molecules, resulting in improved patient compliance and better long-term clinical control of the disease. Nanocarriers can incorporate different polymers to increase ocular drug bioavailability. For example, cellulose derivatives are commonly used as viscosity enhancers to increase the retention time of drugs in the tear film without causing blurred vision. Additionally, mucoadhesive polymers can electrostatically bind to negatively charged mucin and reduce their lacrimal clearance. Nanocarrier DDS can transverse the ocular barriers to reach the target site. Some permeation enhancers enable the nanocarriers to open tight junctions between epithelial cells temporarily, while others form highly soluble host-guest complexes that increase hydrophobic and hydrophilic membrane permeability. To prevent the non-specific distribution of ocular drugs, nanocarrier platforms have incorporated targeting moieties to direct their delivery. Moreover, targeted drug-delivery strategies can be responsive to a disease-specific stimulus, which prevents them from releasing the drugs in non-target tissues. Finally, nanocarrier delivery systems can increase the stability of drugs in the eye by reducing their interactions with tear proteins, and also reduce their electrostatic modification by the different pH levels in different eye structures.

### 2.2. Ideal Characteristics of Nanocarriers

Ideally, nanoparticles are designed to have the following properties (Figure 1):(1)Uniform nano-size distribution.(2)High stability in biological media.(3)High payload capacity.(4)Advanced bioavailability.(5)Biodegradability and high biocompatibility.(6)Controlled release.(7)Controlled cellular uptake and improved pharmacodynamic effect.(8)Ease of upscaling to bulk manufacturing.(9)Cost-effectiveness.

When synthesizing drug-delivery systems, it is challenging to develop a system with all the desired properties for application in different diseases. A key strategy for obtaining such nanocarriers with optimized physiochemical properties (such as aggregation, corona surface charge, solubility, etc.) is combining multiple polymers with nano assemblies. Using copolymers has proven to be more advantageous for nanocarrier DDS synthesis compared to homopolymers, proteins, or lipid-based carriers due to their highly tunable association properties, lower toxicity, and ability to functionalize.

Many chemical strategies are used to blend polymers into miscible nanostructures. Supramolecular assemblies can be formed by activating functional groups on one polymer and controlling their interaction with other host-guest complexing polymers. This results in the formation of nanocarriers with sponge-like morphologies that can non-covalently interact with guest drug molecules. Using cross-linkers, such as dihydrides, genipin, crown ethers, boronic acids, etc., to hyperlink polymers is an important strategy to form hydrogels. Block copolymerization is a common approach for nanomicelle and nanocapsule synthesis. It usually relies on self-assembly by the unfavorable mixing enthalpy of the polymers in the solution. It is important to note that the morphologies (sphere, rod, cylinder, gyroid, and lamella) direct the properties of the nanocarriers. Therefore, maximizing the effectiveness of nanocarriers as drug-delivery systems is crucial to managing their creation.

### 2.3. The Role of Polymer Biodegradability and Morphology in Nanocarrier Drug Release

The biodegradability of the polymers used in nanocarrier DDS results in biocompatible metabolites and prevents the adverse effects accompanying surgical removal of non-biodegradable DDS. The morphology of the biodegradable DDS plays an important role in polymer design given its effect on drug release, cell uptake, and diffusion in cell media [1]. Firstly, DDS can have hydrophilic pores that allow water diffusion and displacement of the drug. Secondly, the use of certain polymers [2] allows for surface erosion of the nano assemblies while protecting water-vulnerable drugs. Similarly, bulk-erosion can occur when the entire scaffold is degraded following the rapid water flow into the system. This type of burst release is sub-optimal in drug-delivery applications, as it can result in an unpredictable release rate and pattern. Therefore, assessing the biodegradability of polymers is essential to optimize the characteristics of proposed drug-delivery systems such as their drug release, drug loading, and cellular uptake. 

### 2.4. The Different Biodegradable Polymers and Their Advantage

The use of biocompatible and biodegradable polymers in ocular drug delivery is becoming increasingly popular. Surface properties such as size and charge can affect the binding of these polymers and differ with each region of the eye. Below are the biopolymers used in ocular drug delivery. 

#### 2.4.1. Hyaluronic Acid

Hyaluronic acid is an anionic polymer and one of the main components of the Eukaryotic extracellular matrix. Given their negatively charged surface, this polymer can permeate the retina by not aggregating in the vitreous humor which also contains negatively charged proteins [3]. Hyaluronic acid can retain water which makes it highly suitable for hydrogel formulations. It is used in most preservative-free artificial tear formulations, allowing prolonged tear film stabilization because of its high water-retention capacity. The addition of crosslinking agents or the activation of Hyaluronic acid by adding functional groups allows for the synthesis of hydrogels with increased mechanical strength and enhanced host-guest groups. Furthermore, Hyaluronic acid can be used to coat liposomes to elongate their drug release and increase their cellular uptake. These formulations are highly used in ocular drug delivery [4]. Hyaluronic acid-based nanocomposite hydrogels have been constructed to load drugs contained in liposomes. This strategy enables combining the slow-release profiles of liposomes with the water-driven release of hydrogels at the specific site of action.

#### 2.4.2. Cellulose 

Cellulose nanocrystals are stable colloidal dispersions that are rod-shaped and extracted from cellulose fibers. Cellulose-based liquid crystals can self-assemble into nanorods, nanospheres, nanosponges, and nanorods upon functionalization with a copolymer and have shown great potential for ocular drug delivery [5]. Hairy nanocelluloses have been produced by periodate oxidation to solubilize the amorphous regions of crystalline nanocellulose and, at the same time, allow for the cleavage of cellulose chains in the amorphous regions. This results in nanorods with soluble amorphous regions attached to dialdehyde-modified cellulose chains on both ends. Therefore, these structures are highly versatile and have an important role in improving the viscosity of several conventional eye drops. The different cellulose derivatives have been applied extensively in ocular drug [6,7] delivery and they have the advantage of the ease of bulk manufacturing [8]. Carboxymethylcellulose sodium is a cellulose-derived polymer that is currently used in formulations to treat dry eyes. It is important to use in gels due to its ability to form transparent gels when dispersed in water. Finally, Hydroxypropyl methylcellulose (HPMC) is a non-ionic cellulose derivative that has high miscibility when combined with hydrophobic or hydrophilic polymers. HPMCs swell when in contact with water, resulting in the formation of hydrogels with enhanced mucoadhesive properties. HPMCs are used as the base of many polymeric DDS in ocular drug delivery [9]. 

#### 2.4.3. Chitosan 

Chitosan is derived from Chitin, which is commonly found in the exoskeletons of many animals. Chitosan requires chemical modification such as acylation, hydroxylation, or esterification to solubilize them in aqueous solutions. Chitosan derivatives are mucoadhesive and possess unique in situ gelling properties, making them suitable for ocular drug-delivery applications with a high potential for advancing glaucoma DDS [10]. It is a cationic polymer that reacts with the negatively charged mucins within the conjunctiva, therefore having excellent corneal permeation that may accumulate in the anterior eye. The unique ability of chitosan to increase drug permeability by temporarily opening tight junctions has been demonstrated by researchers [11]. The functionalization of the ammonium functional groups on chitosan to produce quaternary ammoniums increases the antibacterial activity of chitosan, making them better candidates for the delivery of many gram-positive eye diseases [12]. 

#### 2.4.4. Alginate

Alginate is an anionic copolymer composed of an M unit of β-d-mannuronic acid and a G unit of α-l-guluronic acid units conjugated into a polymer with different GG, MM, and GM proportions. Alginate and chitosan are commonly used in ocular drug delivery as copolymeric nanoparticles since the functional groups on the chitosan derivatives can interact with the eye membranes and can enhance the encapsulation efficiency of alginate [13]. Alginate has the potential to exert cell immobilization, which makes it a suitable polymer in hydrogels. Advances in nano chemistry research have resulted in terminable hydrogels due to the reversible gelation of alginate [14].

#### 2.4.5. PLGA

Poly(lactide-co-glycolic acid) is among the most common polymers used in drug delivery since they are subject to abundant modifications. Their size and surface potential can be enhanced, which selectively alters their membrane permeability. Jiang et al. compared several commonly used PLGA copolymeric nanoparticles and identified that copolymers consisting of hydroxypropyl-β-cyclodextrin and PLGA had the highest entrapment efficiency and trans-ocular permeation [15]. PLGA can also be modified with the use of polyethylene glycol (PEG), to construct biocompatible nanoparticles with sustained drug release and high entrapment efficiency profiles. Studies have also shown that topical administration of PLGA nanoparticles can deliver the guest drug to the posterior segment of the eye [16]. Poly(α-L-glutamic acid)(PGA) is another type of synthetic polymer that can also be conjugated with PEG to form nanoparticles with a dense hydrophilic shell [17]. These polymers have recently been evaluated in clinical trials [17]. 

#### 2.4.6. Poloxamers

Poloxamers are a class of biodegradable, mucomimetic, non-ionic surfactants that are made up of a central block of polyoxyethylene molecules surrounded by two outer blocks of polyoxypropylene molecules. Poloxamer 407 has been approved by FDA as a vehicle for ocular drug delivery and has been used in various ocular formulations such as eye drops, gels, and solutions.

#### 2.4.7. Cyclodextrins 

Cyclodextrins are cyclic oligosaccharides that consist of glucopyranose units which can form hydrophobic cavities with externally hydrophilic surfaces. Cyclodextrin nanoparticles for drug delivery have been shown to increase the bioavailability of many guests, primarily due to the advantage of using the guest-host chemistry of isolated for unique complexation of different guests. Functionalizing the alcohol groups on the hydrophilic surfaces of cyclodextrins with several molecules can enable their crosslinking and is known to enhance their physiochemical properties. A wide range of cyclodextrin-based formulations has been shown to have great potential for ocular drug delivery [18].

### 2.5. Types of Nano-Based Drug-Delivery Systems: Characteristics and Enhancements

Nano-based DDS have been designed to improve sustained and targeted drug delivery to various targets within the eye. The characteristics of these systems are outlined in Figure 2.

#### 2.5.1. Nanomicelles

Nanomicelles are spherical structures formed by surfactant molecules that self-assemble in polar solvents such as water. With the hydrophobic tails of the surfactant pointing inward and the hydrophilic heads facing outward, nanomicelles are ideal for encapsulating hydrophobic compounds and targeted delivery to specific locations in the body. Higher encapsulation capacity is achieved by increasing the concentration of amphiphilic polymers, while polymeric micelles exhibit a lower critical micelle concentration and a more stable shape. Enhancing ocular micro adhesion through the addition of mucin-targeting moieties (such as cyclic peptide ligand and phenylboronic acid) has been demonstrated [19]. The shrinking of nanomicelles [20] has also been shown to increase their efficiency, though their premature degradation in systemic administration remains a barrier to clinical translation [21]. Crosslinking polymeric nanomicelles can enhance stability to avoid premature drug release [22] and enable stimulus-responsive release in topical ocular administration [23].

Nanomicelles have been developed not only to enhance corneal penetration but also to target the posterior segments of the eye. In vivo studies have demonstrated the efficacy of chitosan oligosaccharide-valylvaline-stearic acid nanomicelles in reaching the posterior segments through the conjunctival route [24].

#### 2.5.2. Liposomes

Liposomes are a commonly used formulation in ocular drug delivery, known for their advantageous precorneal penetration and rapid conjunctival uptake [25,26]. These vesicles are composed of one or more phospholipid bilayers that self-assemble into spherical vesicles with a hydrophilic core in aqueous media. Liposomes are highly flexible for chemical modification to fit the physiochemical properties of the drug and the microenvironment of the target. For instance, altering the head group charge using positively charged stearyl amine or negatively charged diacetyl phosphate has been shown to selectively modify the liposome-mucin and corneal permeability interaction [27]. To enhance their stability and suitability for certain applications, PAMAM-coated compound liposomes are commonly used, as the synthetic polymer coat is biocompatible and has been shown to increase the bioactivity of the drug by 1.6 times compared to normal liposomal formulations [25]. 

#### 2.5.3. Dispersed Nanoparticles

Nanoparticles have gained significant attention as a strategy for solubilizing host molecules due to their ability to form supramolecular assemblies. These assemblies are made up of molecular subunits held together by various forces such as van der Waals forces, hydrophobic interactions, electrostatic interactions, hydrogen bonding, and cation–π interactions [28].

The reduced size and lack of aggregation in dispersed nanoparticles increase their carrying efficiency, enabling them to transport a greater number of drugs to the target tissue [28]. Additionally, their mucoadhesive characteristic, enhanced permeability, and retention effect allow nanoparticles to accumulate in the target tissue, making them highly attractive for commercial ocular applications [29,30].

Nanoparticles can be designed with various self-assembling topologies, such as nanospheres, nanorod assemblies [31], and nanocapsules, to encapsulate guest drugs. Their structure is highly dependent on the nature of polymers used to develop the DDS since its formation is affected by the shape of the template polymer, solvent type, surface charge of polymer, type of cross-linkers used, type of polymer blending reaction, and purification method. The ability of these polymers to respond rapidly to changes in the microenvironment and release guest molecules in a controlled manner makes them efficient drug-delivery carriers. Nanorod formulations, in particular, exhibit improved drug delivery properties for ocular applications, as they show a lower rate of uptake by macrophages and longer half-life compared to nanospheres [32,33]. 

Nanocapsules offer a promising solution for ocular drug delivery. Their core consisting of the drug and the protective polymer coating enables them to penetrate the eye’s barriers effectively and reach the target tissue. This enhances the drug’s efficacy and extends its residence time in the eye. The versatility of nanocapsules has been demonstrated in the successful delivery of various drugs, including anti-inflammatory and anti-glaucoma agents [34]. With their ability to protect drugs from degradation and improve their therapeutic effects, nanocapsules continue to attract attention as a valuable tool in ocular drug delivery.

Solid Lipid Nanoparticles (SLNPs) are a promising drug-delivery system composed of a solid lipid core surrounded by a stabilizing surfactant. This unique structure offers several advantages over traditional liposomes and microemulsions, including improved stability, increased drug loading capacity, and an easier manufacturing process. SLNPs also show higher thermal stability and longer shelf life compared to other types of nanoparticles [35].

#### 2.5.4. Dendrimers

Dendrimers are a type of repeating multibranched polymers that feature a dense core and functional groups attached to the surface. Dendrimers are formed of radially symmetric and branched molecules that cross-link several spherical layers. Growing outwards, the core interacts with monomers containing one reactive and two dormant groups, which activates a cascade reaction to form peripheral branches. They can encapsulate different drugs in their large central core. Due to their structure, dendrimers exhibit a high encapsulation efficiency and well-defined biodistribution, making them a promising candidate for ocular drug-delivery systems (DDS). Their core-shell architecture and narrow polydispersity allow for the creation of dendrimer hydrogels, which can be adjusted for crosslinking and gel properties by controlling reactant concentration or functional group density on the dendrimer surface. These hydrogels have been shown to have large internal hydrophobic pores and high loading capacity, making them a valuable tool in ocular drug delivery [25,36,37].

#### 2.5.5. Hydrogels

Hydrogels are attracting attention for their potential in ocular drug delivery. Various chemical and physical crosslinking between copolymers allow for the formation of molecules with large water-accommodating pores. The surface polarity and solubility of the chosen polymers typically influence the morphology of the pores. To match the body’s physiological needs, polyacrylic acid is used in combination with pH-responsive polymers that allow for controlled and sustained drug release at a pH of 5.5. Different chemical modifications, such as amine, ortho ester, imine, and hydrazone, can alter the pH sensitivity of these hydrogels. Other polymers, such as cellulose, chitosan, N-isopropylacrylamide, poloxamers, PLGAs, and PEGs, are used to create thermo-responsive hydrogels that are injectable at room temperature and turn into gels at higher temperatures [38]. Ultrasound-responsive hydrogels can also be used to help drugs penetrate the eye’s barriers and prevent off-target release [39].

The extended drug release profile of hydrogels reduces the frequency of intravitreal injections and allows for targeted delivery to the site of action in the posterior segments of the eye. By incorporating targeted moieties and drugs into the hydrogel in its sol state, the risks associated with frequent injections are minimized.

#### 2.5.6. Nanosuspensions and Nanoemulsions

Nanosuspensions and nanoemulsions have been widely accepted as DDS for ocular drug delivery. According to Nagai and Otake (2022), they can effectively deliver poorly soluble and permeable drugs. Among nano-based biodegradable DDS, nanoemulsions have been widely approved for ocular drug delivery. Nanoemulsions are small droplets of one liquid dispersed in another liquid, which can improve the solubility and stability of poorly soluble drugs and make them suitable for ocular administration. The use of amphiphilic salts of cholesterol in nanoemulsions has been found to greatly stabilize the formulations and make them effective for delivering antiglaucoma, anti-inflammatory, and antiviral drugs [40]. By increasing the fraction of the dispersed oil phase, the viscosity of nanoemulsions can be increased to make them more bioavailable at their target site. Additionally, the addition of water-soluble polymers can allow for the formation of a gel with a long retention time.

Nanosuspensions, on the other hand, are aqueous dispersions of insoluble drug particles stabilized by surfactants. They are especially useful for delivering drugs with large molecular weights and high melting points or drugs that are unable to form salts [41]. Nanosuspensions are advantageous as they can avoid high osmolarity produced by ophthalmic solutions while prolonging drug release. Although they are often prepared in an aqueous medium, their chemical stability may be affected by hydrolysis. The use of other polymers to stabilize nanosuspensions can prevent their aggregation and provide physical stability, including poloxamers, phospholipids, and cellulose derivatives [42].

#### 2.5.7. Microneedles

Microneedles have shown their potential to deliver ocular drugs to specific eye segments, such as the subarachnoid and posterior regions. This method offers several benefits compared to subconjunctival injections, such as improved accuracy in target delivery, ease of self-administration, and reduced risk of complications that can affect the integrity of the eye barriers. Furthermore, by using hollow microneedles, higher concentrations of drug molecules can be delivered, which can be controlled by adjusting the copolymer grafting ratios [43].

## 3. Anatomical and Physiological Factors Affecting Ocular Drug Delivery

Ocular drugs can be administered via several routes, including topical, subconjunctival periocular, intravitreal, and systemic delivery. The three main routes for delivering drugs to the back of the eye are topical, systemic, and intravitreal. Topical application is the simplest method and is commonly used, but only about 5% of the applied dose can penetrate the internal structures of the eye due to barriers such as the tear film, cornea, conjunctiva, vitreous, blood–aqueous barrier, and blood–retina barrier [44] (pp. 27–36). (Figure 3 anatomical and physiological factors affecting ocular drug delivery) The tear film has a complex structure of three layers (lipid, aqueous, and mucin) and is constantly removed from the eye surface by the lacrimal fluid secretion, leading to the rapid removal of the drug. A significant portion of a drug instilled into the eye is eliminated through the nasolacrimal duct, which contributes to undesirable systemic absorption of the drug and reduces its availability for the targeted ocular tissue. The cornea serves as a mechanical and chemical barrier, limiting the access of exogenous substances into the eye and protecting the intraocular tissues. It is a semi-permeable membrane, allowing passive transfer of material across its cells, but tight junctions on the surface of the corneal epithelium prevent diffusion of macromolecular and hydrophilic molecules. Intravitreal administration provides a direct path to the vitreous and retina, but diffusion of larger and positively charged drugs across the RPE barrier to the choroid may be impeded. The elimination of drugs from the aqueous humor happens through aqueous turnover and venous blood flow from the anterior uvea. Lastly, the blood–ocular barrier (BOB) is a major impediment to systemic and topical drug delivery in the eye’s anterior and posterior chambers. It is composed of the blood–aqueous barrier (BAB) and the blood–retinal barrier (BRB). The BAB, related to the anterior chamber, consists of endothelial cells, iris, ciliary muscle, and pigmented and nonpigmented epithelium cells, characterized by tight junctions that restrict drug molecule entry. In eyes with uveitis or uveitic glaucoma, the breakdown of the BAB from dilation of the iris and ciliary body vessels allows inflammatory cells, mediators, and proteins to enter the usually immune-privileged intraocular environment [44] (pp. 27–36).

## 4. Nano-Based DDS for Glaucoma

Glaucoma is a group of eye diseases that damage the optic nerve, leading to progressive vision loss and blindness. It is the leading cause of irreversible blindness globally. To this day, intraocular pressure (IOP) management is the cornerstone of treatment for glaucoma. With the aim of reducing intraocular pressure (IOP), there has been a recent surge in the use of nanocarriers to enhance ocular drug delivery. This development is particularly significant as the long-term use of conventional IOP-lowering agents, which are based on poorly penetrating molecules, can cause frequent ocular toxicity and intolerance due to their adverse effects on the corneal epithelium [45].

### 4.1. Nano-Based DDS Based on Regulating Intraocular Pressure (IOP)

#### 4.1.1. Preclinical Studies

One advantage of nanoparticles is the ability to modify them with biodegradable polymers, which can enhance the bioavailability of the drug they are delivering. This is especially important for glaucoma drugs such as brinzolamide, which has low solubility and thereby limits its therapeutic efficacy. Various studies have formulated PLGA-modified (Polylactic-co-glycolic acid) nanoparticles to have a sustained release of brinzolamide that significantly reduced IOP in animal models with minimal toxicity [15,46,47]. Song et al. (2020) used a coaxial electrospray technique to coat PLGA nanoparticles with phosphatidylserine (PS), improving entrapment efficiency and corneal permeation with topical delivery of brinzolamide. However, some systemic absorption was observed where there was an IOP reduction in the untreated eye as well [46]. Salama et al. (2017) used subconjunctival injection to enhance the prolonged release of these nanoparticles and provide a more targeted drug delivery where they found efficacy was dependent on nanoparticle size. PLGA nanoparticles were also used to encapsulate a hybrid compound SA-2, a nitric oxide (NO) donor and superoxide dismutase (SOD) activator [48]. A single-dose slow release of SA-2 significantly lowered IOP for up to 72 h. Furthermore, increasing SOD enzyme activity provided cytoprotection in the human trabecular meshwork (TM) cells.

Other polymers recently used for forming nanoparticles are γ-cyclodextrin [49] and chitosan [50], and a natural organic polymer based on galactomannans [51]. Lorenzo-Soler et al. (2020) used cyclodextrin nanoparticles to administer angiotensin receptor blockers, candesartan, and irbesartan, preventing the conventional side effects observed with oral administration, while effectively lowering IOP comparable to timolol eye drops. Although promising, further pharmacokinetic profiling is required since individual variability in IOP was found in their rabbit animal model [49]. In a study by Barwar et al. (2019), nanobrimodine was delivered with ultra-small chitosan nanoparticles for open-angle glaucoma, a well-known biodegradable polymer with a high capacity for drug loading and slow release. Given the size (28 ± 5 nm), this was able to enter through receptor-mediated endocytosis seen through in vitro studies [50]. Mittal et al. (2019), formulated a bio-adhesive, non-toxic polymeric nanoparticle using galactomannans for the delivery of dorzolamide hydrochloride. A prolonged release improved IOP-lowering effect compared to conventional eye drops. An advantage of this formulation is the safe and economic extraction compared to synthetic polymers [51]. Finally, through intracameral injection, Tan et al. (2021) delivered miRNA through polydopamine-polyethylenimine (PDA/PEI) Nanoparticles. The drug-delivery system improved penetration and stability, allowing for comparable transfection efficacy with lower cytotoxicity, and was also effective in lowering IOP in vivo [52].

Although Mesoporous Silica Nanoparticles (MSNs) have poor degradability given a pure silica framework, some studies have recently modified MSNs to achieve biodegradation suitable for clinical application. Fan et al. (2021) used biodegradable hollow mesoporous organosilica (HOS) nanocapsules to deliver NO in a stimulus-responsive manner for improved penetration and bioavailability [53]. Although it showed an IOP-lowering effect, prolonged use of NO-MSNs was previously found to increase IOP elevation following the initial decrease [54]. It was suggested that this was caused by outflow tissue damage through protein nitration, and co-delivery of antioxidants such as MnTMPyP may ameliorate these side effects of prolonged use. Alternative methods to develop biodegradable hollow polymeric nanocapsules are currently being extensively researched given their high drug-loading capacity and ability to better control drug release [55].

Niosomes are lipid-based vesicular systems, providing improved residence time and corneal permeation as a DDS. Zafar et al. (2021) created a chitosan-coated niosome, improving biodegradability and bioadhesion, to deliver carteolol (CT) [56]. Similarly, using niosomal gels, pilocarpine, and latanoprost were successfully delivered to reduce IOP in in vivo rabbit models [57,58]. However, niosomes are known to have a limited shelf life and poor drug entrapment efficiency. To optimize this, proniosomes (gel or granular) can convert to niosomes rapidly upon hydration, minimizing niosomal dispersions, and providing higher physical stability [59,60].

Cubosomes are formed by a continuous lipid bilayer, similar to the structure of cell membranes of the corneal epithelium. It has therefore been investigated as a possible DDS for ocular delivery. Teba et al. (2021) used a novel cubosome system to deliver acetazolamide, with improved corneal permeation, ocular residence time, and no signs of ocular irritation [61]. Similarly, Huang et al. (2017) developed a timolol maleate-delivering cubosome system. Both revealed improved bioavailability of either drug as well as an improved IOP-lowering effect [62]. In both studies, the cubosome was developed with biodegradable polymers poloxamer 407 (P407), and GMO (glyceryl monooleate) [61,62].

Nanoemulsions are nanoscale droplets capable of carrying both hydrophilic and hydrophobic materials. They have been reported to be feasible carriers of travoprost and brinzolamide, enhancing drug absorption and prolonging IOP-lowering effects [63,64]. However, the synthesis of nanoemulsions requires a high level of surfactants, resulting in a high risk of cytotoxicity, which can be further exacerbated with the addition of preservatives [64]. Long-term safety tests regarding toxicity are required for use of this DDS.

To improve treatment adherence and decrease the frequency of administration, optimizing ocular inserts using nano-based carriers has been a common focus of research in ocular pharmacology. Several studies have used chitosan, a highly biodegradable polymer, to develop ocular inserts and films for topical delivery. Modifications were carried out to improve the solubility of chitosan, such as the addition of chondroitin sulfate and hydroxyethyl cellulose [65,66]; these modified chitosan inserts were able to lower IOP by delivering 4-aminodiphenylamine (a benzamidine derivative) and dorzolamide, respectively. Interestingly, with a sustained release, both drugs also exhibited a neuroprotective effect towards the retinal ganglion cells (RGC). Li et al. (2020) presented an eco-friendly method of producing the chitosan polymer dissolved in a water-based film, overcoming the solubility issue while retaining high cornea permeability [67]. Other biodegradable polymers reported are sodium alginate-ethyl cellulose inserts to carry hydrophilic drugs [68], as well as a cyclodextrin multilayer film incorporating polybutyl acrylate-co-ethylene oxide (PBAE) and graphene oxide, which enabled a time-controlled release of brimonidine, dependent on layer thickness in vitro [69].

Nanomicelles have been used to co-deliver latanoprost and timolol through a drug-laden contact lens developed by Xu et al. (2019) [70]. The authors used biodegradable mPEG-PLA nanomicelles prepared through thin-film hydration, where the ultra-small particle size ensured transparency and transmittance of the embedded contact lens. Although this DDS provided a slow, sustained release of both drugs (over 120–144 h) and confirmed ocular safety, it was reported that this system may affect the physical properties of the lens, becoming rough after drug release. Another co-delivery system was provided by Samy et al. (2019), where they developed PCL thin-film implants to deliver timolol and brimonidine [71]. This DDS provided a controlled release of both drugs; however, systemic absorption and related side effects were not measured. These polymeric films were also used to deliver a novel hypotensive agent, DE-117, which provided a sustained IOP-lowering effect, but given the bulky size of the implant, it created a high risk of device migration and corneal endothelium damage [72].

Hydrogels have been extensively researched as a promising ocular delivery system, given their properties of in situ sol-to-gel formation triggered by environmental factors such as pH, temperature, etc. [73]. However, to ensure the sustained release of the drug, recent studies have been developing hybrid systems, embedding hydrogels with biodegradable nanoparticles. This has led to a successful DDS with sustained release of brimonidine tartrate [74], timolol maleate [75,76], and bimatoprost [77] and co-delivery of curcumin nanoparticles and latanoprost [78] from a thermosensitive in situ hydrogel. In another example of DDS based on dual functions, Chou et al. (2017) presented the delivery of pilocarpine-loaded with antioxidants GA (gallic acid) through a biodegradable gelatin-based thermogel [79]. Along with providing antioxidative effects, GA was able to accommodate the controlled release of pilocarpine from the thermogel. The degradation depended on the redox radical initiation reaction temperatures (20–50 °C); therefore, if modulated, it can provide a sustained release (found ideal at 30 °C). This further opens the discussion on modifying biodegradability to improve the sustained release of the drug. Luo et al. (2019) found that an increase in the amination degree of gelatin in biodegradable thermogels enhances resistance to biodegradation [80]. In another study by the same group, it was presented that increasing the deacetylation degree enhances resistance to biodegradation in chitosan-based thermogels [81].

Liposomes can carry both hydrophilic and hydrophobic drugs and are highly biodegradable nanocarriers. Jin et al. (2018) presented a TPGS-modified (tocopheryl polyethylene glycol succinate) liposome carrier for brinzolamide with a greater sustained release and maintained IOP reduction, with no significant side effects [82]. However, a common drawback to using liposomes is their low stability, low entrapment efficiency, and rapid release of hydrophilic drugs [83,84]. To optimize this, Hathout et al. (2018) developed gelatinized core liposomes for the sustained release of the hydrophilic drug, timolol maleate, which significantly increased entrapment efficiency with no signs of ocular irritation.

PAMAM (polyamidoamine) dendrimers have been used for the sustained release of timolol with no signs of cytotoxicity or ocular irritation in vitro. Further studies with in vivo models for safety and pharmacokinetic profiling are required to optimize drug loading as well as investigate the effects of chronic application [85]. Lancina et al. (2017) used electrospun dendrimer-based nanofiber mats to deliver brimonidine tartrate (BT) [86]. This method improved the efficacy of IOP lowering with daily dosing over 3 weeks, but not with a single dosage.

Other notable DDS systems include self-assembly drug nanostructures (SADN) [87] and phase transition microemulsions (PMEs) [88] which are both novel systems that aided in sustained IOP-lowering effect. Cytotoxicity and systemic effects remain to be investigated for both delivery systems. Cell-softening agents, such as ligands targeting FLT-4/VEGFR3 receptors, were delivered through modified micelles, improving receptor targeting and lowering IOP [89]. Further improvement in corneal permeability and sustained release is required, given that effects remained up to only 48 h. Nanosuspensions were also reported to deliver acetazolamide by Donia et al. (2020), where they demonstrated sustained drug release, improved acetazolamide solubility, and thereby improved bioavailability [90]. Hyaluronic acid (HA) was used to stabilize the nanosuspension; however, this modification was only able to maintain dispersion characteristics for up to 6 months. Therefore, the common limitation of improving the stability of nanosuspensions remains to be further investigated.

Interestingly, a drug-free, non-surgical method of reducing IOP was proposed by Chae et al. (2020), where they use a hyaluronic acid hydrogel microneedle injection to expand the suprachoroidal space [91]. This method creates a temporary passage between the anterior chamber and the suprachoroidal space, similar in principle to suprachoroidal minimally invasive glaucoma surgeries (MIGS) such as Cypass and iStent Supra, but without the associated loss of endothelial cells. This facilitates aqueous humor drainage through the uveoscleral outflow pathway. They were able to extend the lowering of IOP without drugs or surgical intervention, with no significant complications. Further mechanistic studies are required to validate this method, and there is a need to address preventing fibrosis with multiple injection treatments.

Several studies have recently modified existing DDS such as microspheres [92,93], solid lipid nanoparticles (SLNs) [94], and chitosan nanoparticles by embedding montmorillonite (Mt) [95] in the biodegradable hybrid polymer. Mt, a silicate with high biocompatibility, has an overall negative surface charge and can create an ion complex with cationic drugs such as betaxolol hydrochloride (BH) and brimonidine. This enables a controlled, sustained release of the drug with a hybrid nanocarrier, shown by a prolonged decrease in IOP.

Although solid formulations (gels, contact lenses, implants, etc.) for ocular delivery are beneficial in terms of prolonged ocular residence, there is a need to avoid having them interfere with patients’ vision and improve comfort. Recent DDS studies propose the use of electrospinning biodegradable polymers that are more adaptable and rapidly dissolve in the cornea due to their highly porous structure. Andreadis et al. (2022) created an in situ electrospun film gel for the delivery of timolol maleate [96], and similarly, Morais et al. (2021) developed electrospun ocular implants for acetazolamide delivery [97]. Both showed a significant sustained IOP-lowering effect and increased local delivery. However, further optimization of the sterilization methods for the implant is required, to avoid infection and adverse events without compromising the efficacy of the polymer.

#### 4.1.2. Clinical Studies

To improve sustained release while limiting systemic exposure, Rubiao et al. (2021) conducted a Phase 2 controlled study comparing a bimatoprost topical chitosan-based insert to conventional bimatoprost eyedrops (Lumigan^TM^). Notably, 16 patients with Primary Open Angle Glaucoma (POAG) and 13 patients with ocular hypertension were enrolled; however, the control group had a small sample size of 5. Given the biodegradability of chitosan, the insert did not need to be removed from the patients afterwards. This was a safety study, where the insert did not cause any major side effects, visual acuity changes, or central corneal thickness. The insert provided an IOP reduction of 30% by the third week, compared to the 35% reduction with eye drops. This study confirmed a better therapeutic regime, changing from daily dosage to using inserts in 3-week intervals, improving patient compliance [98]. Their next reported step is to conduct Phase 3 confirmatory studies.

FDA-approved in 2020, Durysta^TM^ is a bimatoprost implant composed of poly-lactic acid and poly-lactic-co-glycolic polymers (similar to biodegradable sutures used in clinical settings) [99]. It is placed intracamerally intended to provide a non-pulsatile, sustained release of 10 μg bimatoprost. It is currently undergoing multiple Phase 3 trials in patients with open-angle glaucoma (OAG) or ocular hypertension (OH) regarding long-term safety and efficacy [100]. It has been found non-inferior to timolol eye drops, improving adherence, and reducing the treatment burden associated with glaucoma [101]. There is a potential risk of corneal adverse events, intraocular inflammation, or endophthalmitis. Following up on their Phase 1/2 patients [102,103], implant biodegradation was variable among patients, with clinically significant degradation by 12 months; however, the IOP-lowering effect was maintained. This implant is a viable option for patients with glaucoma who are unreliable or have dementia, and who are poor candidates for incisional glaucoma surgery.

The sustained release of bimatoprost was also investigated through an ocular ring in double-masked randomized Phase 2 clinical trials by Brandt et al. (2017) [104]. The long-term efficacy (7- and 13-months post-exposure) was studied in 65 and 63 patients, respectively. It was shown that bimatoprost ocular rings were safe and well tolerated, however, with the side effect of mucus discharge. It was suggested that clinically relevant IOP reduction can be obtained with applications at 6 month-intervals. Given that the ocular ring is formed by a silicone matrix over a polypropylene structure, studies regarding its biodegradability may elucidate areas of improvement in terms of biocompatibility.

A randomized controlled trial was conducted by Kouchak et al. (2017) on patients with OAG and OH to investigate the effects of dorzolamide nanoliposomes (DRZ-nanoliposome) on IOP [105]. Twenty patients were enrolled and the DRZ-nanoliposomes’ efficacy and safety were compared to a control group receiving dorzolamide eye drops (Biosopt^TM^). Assessed at days 14 and 28, a greater reduction in IOP was found in the DRZ-nanoliposomes group with extended release and higher intensity postulated to be due to increased adhesion and corneal permeation with the liposome delivery system. Any irritation or redness reported was similar between both groups, commonly caused by dorzolamide itself. This can possibly improve treatment outcomes by reducing the frequency of administration without reducing efficacy.

Table 1 presents a summary of the studies referenced in this section.

### 4.2. Nano-Based DDS for Neuroprotection

Although conventional therapeutic strategies towards treating glaucoma are decreasing IOP, this remains inadequate in addressing the permanent damage to retinal ganglion cells (RGCs) and the optic nerve in advanced diseases. There are also forms of normotensive glaucoma, and cases of controlled IOP where visual loss progresses [113]. Therefore, there are neuroprotective strategies being investigated for glaucoma. Nano-based biodegradable DDS are a promising approach for these therapies given that effective treatment will require long-lasting and increased drug permeation to deliver these drugs to the posterior segment of the eye.

#### 4.2.1. Preclinical Studies

Beginning with a drug treatment previously discussed, in addition to its clinical use in reducing IOP, brimonidine also exhibits neuroprotective effects by regulating excitatory NMDA receptors in RGCs [114]. To improve its delivery to the posterior eye, two studies were found to investigate its neuroprotective effects in a glaucoma model. Lou et al. (2021) developed a dual-function PDA biodegradable nanoparticle loaded with brimonidine, where PDA (Polydopamine), in addition, aids in ROS scavenging and anti-inflammation effects [115]. Rodrigo et al. (2020) proposed using a biodegradable LAPONITE^TM^ synthetic clay, composed of nanoscale crystals and capable of controlled release of drugs [116]. Both systems achieved sustained and increased permeation of brimonidine, promoting RGC survival. The LAPONITE system had some adverse side effects pertaining to systemic absorption of brimonidine, eliciting CNS depressant effects and thereby early deaths among rats in their in vivo model.

Given that neurotrophic factors cannot cross the blood–retinal barrier and degrade rapidly, some studies investigate the delivery of neurotrophic factors (NFs) to prevent retina damage in glaucoma models. Yang et al. (2021) delivered CTNF and oncostatin M (OSM) successfully with nanoparticles, improving RGC survival and preservation of vision and photoreceptors [115]. Similarly, Giannaccini et al. (2017) delivered NGF and BDNF with biodegradable magnetic nanoparticles for improved localization [117]. It was able to prevent RGC loss with lower dosages of NFs; however, given that they used a zebrafish model that has better optical regeneration, further in vivo studies of animals whose eye anatomies are closer to the human eye anatomy are required. Both studies were able to achieve a sustained release of NFs through their respective methods. Garcia-Caballero et al. (2017) were able to deliver GDNF/Vitamin E through biodegradable PLGA microspheres with up to 6 months of sustained delivery, allowing for effective neuroprotection [118]. However, the optimal dosage remains to be confirmed. PLGA microspheres were also used for the delivery of dexamethasone, melatonin, and coenzymeQ10 [119,120]. This multitherapy reduced RGC loss along with reducing retinal stress of multiple single doses and showed good tolerance in their in vivo model. Ding et al. (2021) used cubosomes for the delivery of LM22A-4, a small NF mimetic [121]. Although only 17% of the polymer targeted the posterior retina, it improved functional outcomes and prevented RGC loss through a gradual, targeted release.

Hydrogels have also been used to provide a sustained release and improve the bioavailability of neuroprotective drugs. The use of biodegradable chitosan thermogel was used to deliver pilocarpine and RGFP966, an HDAC inhibitor known to protect RGCs from apoptosis and optic nerve damage [122]. The thermogel itself provided antioxidant effects by a methoxylation mechanism. This method enhanced myelin growth and reduced RGC loss. Similarly, pilocarpine was co-delivered with the anti-inflammatory agent ascorbic acid in a study by Nguyen et al. (2019) through a PAMAM dendrimer thermogel [123]. This aided in the sustained release of both agents over 80 days, suppressing inflammation and aiding in the regeneration of stromal collagen and retinal laminin.

Cannabinoids have also exhibited a neuroprotective effect; however, poor delivery to the posterior eye limits their therapeutic efficacy. Kabiri et al. (2018), used an HA-MC thermosensitive hydrogel laden with cannabigerolic acid (CBGA) nanoparticles [124]. This method improved bioavailability and corneal permeation as well as reportedly reduced ocular irritancy. Further in vivo studies regarding its efficacy in neuroprotection are required. A derivative of cannabinoids, Δ9-Tetrahydrocannabinol-valine-hemisuccinate, was delivered using solid lipid nanoparticles (SLNs) which reportedly provided neuroprotective as well as IOP-lowering effects [125]. Although it was successful in providing prolonged residence time, further cytotoxicity and safety profiling needs to be elucidated [126].

Various studies have used nanoparticles as their DDS for neuroprotective agents. Memantine, an NMDA antagonist, was loaded on biodegradable PEGylated nanoparticles, reporting enhanced delivery and reduced RGC loss [127]. An administration regime of twice daily for three weeks was used in their in vivo rodent model, with improved drug tolerability and sustained drug release. Used for a non-invasive gene delivery system, Narsineni et al. (2022) used Gemini (PGL) Nanoparticles to deliver peptide-modified CAP-gemini surfactants, a potential Aβ40 aggregation inhibitor [128,129]. Previous studies have shown that RGC neurodegeneration is associated with Aβ40 deposition [130]. These surfactants provided a 10-fold improvement in Aβ40 aggregation inhibition. Further in vivo models regarding the therapeutic efficacy of glaucoma neurodegeneration are required.

Li et al. (2020) used biodegradable PEG-based nanoparticles for the co-delivery of brinzolamide and miR-124, a miRNA known for its anti-inflammatory and neuronal differentiation effects [131]. This co-delivery provided prevention to RGC damage and IOP-lowering effects through its sustained release. No ocular toxicity was observed; however, further mechanistic studies are required to validate this therapy. Zhao et al. (2017) also used a PEG-based nanoparticle system to deliver DHEA, an FDA-approved S1R agonist [132]. These nanoparticles were conjugated with cholera toxin B domain (CTB) for improved targeted delivery to RGCs. This novel delivery system improved targeting and provided effective RGC protection; however, improvement in entrapment to provide a more efficient, sustained release of NPs is required. Silva et al. (2022) used a chitosan and HA nanoparticle system to deliver Epoetin Beta, which has tissue-protective properties [133]. An in vivo model showed sustained release for up to 21 days and showed no local or systemic adverse effects. Further models investigating therapeutic efficacy on glaucoma models are required.

Another notable DDS-delivering neuroprotective agent is a microcrystal proposed by Hsueh et al. (2021), which contained sunitinib, an FDA-approved multi-kinase inhibitor that promotes the survival of RGCs [134]. They achieved a sustained release with therapeutically relevant concentrations (through a pig model) and were found to provide neuroprotection for at least 20 weeks in a rat optic nerve crush model.

#### 4.2.2. Barriers to Clinical Translation

Current therapeutic trials targeting neuroprotection for glaucoma have been finding difficulty in replicating the in vivo efficacy in humans. Primarily, this could be due to the use of acute damage animal models injuring the optic nerve and RGCs, whereas the progression of glaucoma in humans is chronic and prolonged, yielding differing neuronal dysfunction [113]. Current neuroprotective agents also have low bioavailability, chemical instability, as well as local and systemic adverse effects. For example, brimonidine monotherapy had a great number of side effects including hyperemia, hypersensitivity, and ocular discomfort. Stem cell therapy, NMDA antagonist-drugs, and NFs although all promising, may cause undesired systemic absorption and in general low bioavailability to the posterior eye [135]. The current advance of biodegradable nano-based DDS may overcome these limitations, improving bioavailability, targeted delivery to avoid systemic effects, and sustained release for increased patient adherence. As discussed above, many DDS approaches are at the in vitro or preliminary in vivo stages. Hopefully, future studies conducted on animal models to confirm these neuroprotective findings will pave the path for clinical translation.

Table 2 presents a summary of the studies referenced in this section.

### 4.3. Nano-Based DDS after Laser and Surgical Treatment of Glaucoma

#### 4.3.1. Preclinical Studies

Nano-based DDS can also be used to improve anti-fibrotic, anti-inflammatory, and sustained IOP-lowering effects after glaucoma filtration surgeries and laser trabeculoplasty (SLT). As previously mentioned, an FDA-approved bimatoprost implant, Durysta^TM^ was recently used in a beagle in vivo model to investigate whether the sustained release of bimatoprost aided in IOP-lowering post-selective laser trabeculoplasty (SLT) in the study performed by Ghosn et al., 2022 [136]. This study showed that the implant was responsive in lowering IOP in post-SLT eyes as well, with a sustained IOP lowering for 42 weeks following the implant. This was a small sample size study, and there are obvious limitations when translating this to human patients, who will have greater variability in IOP given that they will be on chronic glaucoma medication prior to any surgical treatment.

Another implant considered for use after glaucoma surgery is a collagen matrix implant loaded with bevacizumab and sodium hyaluronate developed by Andres-Guerrero et al. (2021) [137]. This implant uses the anti-VEGF properties of bevacizumab to promote wound healing and reduce bleb failure through mechanical support provided by sodium hyaluronate. This provided anti-scarring and wound healing post-conventional trabeculectomy. Although biodegradable, there is a risk of the collagen matrix inducing inflammation after degradation. This system improved tissue repair in vitro but was not intense enough to achieve clinical significance. Further improvement in DDS for controlled, sustained release may be required.

Mitomycin C (MMC) and 5-Fluorouracil (5-FU) are commonly applied after filtration surgery to prevent scarring. A PLGA film containing MMC, and 5-FU was formulated by Swann et al. (2019). This provided anti-fibrotic treatment post-trabeculectomy, with a sustained release of both drugs in low dosages. Long-term studies to further elucidate the safety of this system are required. Vildanova et al. (2022) developed a biodegradable modified chitosan and HA hydrogel to improve sustained delivery of MMC and 5-FU [138]. Currently, only in vitro studies have been reported, but sustained release of both drugs was found, with MMC having a longer release compared to 5-FU. Qiao et al. (2017) also used a chistosan-modified hydrogel to deliver heparin post-glaucoma surgery [139]. This method was effective in maintaining filtration bleb and lowering IOP for a prolonged time. However, cytotoxicity profiles for the abovementioned studies, especially the long-term effect on corneal limbal stem cells, are required.

The gelatin-based hydrogel was used by Chun et al. (2021) to deliver siRNA (siSPARC) to reduce subconjunctival scarring post-trabeculectomy [140]. siSPARC is found to reduce pro-fibrotic genes and decrease excessive collagen deposition. Furthermore, it shows no cellular toxicity. However, it is easily degradable, which is why a hydrogel may improve ocular delivery. This study showed the hydrogel DDS to be non-toxic, have excellent biocompatibility, and have an effective reduction in scarring. Further optimizing the dosage to ensure the sustained release of siSPARC is required. siSPARC was also delivered with similar anti-fibrotic characteristics using LbL (layer-by-layer) nanoparticles (Seet et al., 2018) [141].

#### 4.3.2. Clinical Studies

The literature search yielded one recent clinical study pertaining to improving drug-delivery post-glaucoma surgery. Johannesson et al. (2020) used dexamethasone nanoparticles (DexNP) to deliver MMC post-trabeculectomy in a randomized, single-masked clinical trial [142]. With a total of 20 patients, using DexNP proved non-inferior compared to conventional MMC Maxidex^TM^ eye drops. Some limitations include their small sample size and patients being unmasked toward treatment type. Altogether, given the risks and complications associated with MMC administration, this nanoparticle system was able to localize and limit the exposure within the eye, providing a potentially safer alternative to MMC post-glaucoma surgery.

Table 3 presents a summary of the studies referenced in this section.

## 5. Nano-Based DDS for Anterior Segment Diseases

### 5.1. DDS for Ocular Surface Disease

#### 5.1.1. DDS for Dry Eye Disease

Dry eye disease (DED), also known as keratoconjunctivitis sicca, is a multifactorial condition characterized by insufficient or poor-quality tears that result in discomfort, visual disturbances, and instability of the tear film. This can lead to inflammation of the eye surface and cause damage to the ocular surface. It is often accompanied by an increased osmolarity in the tear film. Dry eye disease can be caused by a number of factors, including aging, certain medical conditions such as Sjogren’s syndrome, medications, environmental factors, lifestyle habits, and hormonal changes. Pharmacologic therapies for DED can range from mild cases, such as artificial tears and ointments, to more severe cases requiring topical corticosteroids, immunosuppressants, and autologous tear therapies. The pharmacological treatment options for dry eye disease (DED) vary depending on the severity of the condition. Mild cases may be managed with artificial tears and ointments, while more severe cases may require the use of topical steroids, immunosuppressants, or autologous tear therapy. Each of these treatments has its own limitations and drawbacks. For example, artificial tears need to be applied multiple times per day and rely on the compliance of the patient. Long-term use of topical steroids can cause side effects such as increased intraocular pressure and cataracts, and autologous tear therapy is both costly and involves multiple visits to a healthcare setting for blood draws. Moreover, DED itself can affect ocular drug delivery by reducing the residence time of topically applied drugs and increasing the risk of systemic absorption. DED can also increase the rate of tear turnover which can further reduce the efficacy of topical medications. To maintain a healthy ocular surface and improve drug delivery, other therapeutic strategies have been developed.

Currently, nanoemulsions encapsulating cyclosporine A such as Cyclokat^TM^ and Restasis^TM^ have also been approved for dry eye disease due to their highly solubilized state and their stability improvement. The delivery of the drug to ocular tissue remains hampered by its high molecular weight and its higher affinity with the oil phase of the nanoemulsion. Recently, the use of Cequa^TM^, a nanomicelle solution containing cyclosporine A, has been shown to treat dry eye diseases. After 84 days of treatment, patients have reported increased tear production and improved ocular surface integrity [144]. Cequa^TM^ nanomicelles are amphiphilic surfactants that can increase the dissolution of the hydrophobic cyclosporine A, and its penetration through the tear film and other anatomical barriers of the eye. Recently, KPI-121, a mucin-penetrating particle (MPP) for the delivery of loteprednol etabonate has been approved by the FDA for the treatment of dry eye disease. The nanosuspension (roughly 300 nm) was developed by a milling procedure containing loteprednol etabonate and Pluronic F127 polymer. This delivery system showed a significant advantage over the conventional loteprednol formulation in a New Zealand white rabbit model. KPI-121 achieved a higher ocular exposure with peak concentrations of approximately threefold higher in ocular tissues than the conventional formulation with a single topical delivery [145,146]. These nanoparticles have a low molecular weight and can evade entrapment by mucin and have a reduced clearance rate compared to conventional eye drops. The polymer used is reported to have a longer hydrophobic poly(propylene oxide) chain, that can provide strong hydrophobic absorption on its surface and may further improve drug loading. KPI-121 has shown minimal toxicity in clinical trials [147]. The phase III clinical trial conducted a 2-week course of treatment in approximately 2700 patients and has demonstrated a successful reduction in signs and symptoms of dry eye disease [148]. The F127 polymers form nanomicelle structures and have the ability to form hexagonal morphologies at higher temperatures, which are thought to preserve their overall size and improve the stability of the system [149].

Several novel biodegradable nano-sized DDS have recently been investigated in clinical trials. They aim to reduce the frequent administration times of currently used DDS and to increase the bioavailability of guest drugs at the target site of action. To achieve long-term controlled drug delivery of cyclosporine A, Mun et al. (2019) have synthesized cholesterol-hyaluronate nanomicelles. Crosslinking the nanomicelles using ethylene glycol dimethacrylate and hydroxyethyl methacrylate resulted in the formation of contact lenses that showed a prolonged therapeutic effect on a dry eye disease rabbit model [150]. Novel nanomicelle formulations of cyclosporine A that are based on mPEG-PLA copolymers have been developed. These nanomicelles can be lyophilized and demonstrate increased stability and prolonged shelf life.

Rebamipide has been demonstrated to increase mucin and lipid layers of the tear film and reduce ocular surface dryness. Clinically, rebamipide eye drops have been found to treat tear deficiency and corneal epithelial damage caused by a lack of mucin. Novel drug-delivery systems are being investigated to evade nasolacrimal clearance and increase drug compliance by altering its opaque and turbid appearance. Copolymeric nanoparticles have been developed by Nagai et al., from 2-hydroxypropyl-β-cyclodextrin and methylcellulose and had a more sustained release compared to the commercially available formulation. The use of nanoparticles allows for improved delivery to the goblet cells and meibomian glands, leading to increased stimulation of mucin and lipid production [151]. Another formulation is composed of hydrogenated soybean phospholipids and high-purity cholesterol multilamellar nanoliposomes. They exhibited an equivalent therapeutic effect observed by fluorescein staining compared to the commercially available rebamipide formulation which was several times more concentrated than the nanoliposomes. The nanoliposomes improved drug retention and allowed for sufficient drug concentration at the cornea and aqueous humor, reducing the frequency of administration and the adverse effects [152].

Recently, Wang et al. have reported the synthesis of rapamycin nanospheres based on 3- hydroxybutyrate-co-3-hydroxyvalerate copolymers. Unlike the currently available rapamycin ocular formulations, these nanospheres can penetrate the tear film barrier effectively. They have been shown to increase tear meniscal height, decrease tear break-up time, and improve Schirmer’s test scores, suggesting that they may be advantageous for Sjögren-associated dry eye disease [153]. However, the results of this study are preliminary and further research is needed to determine the long-term safety and efficacy of the nanospheres.

Luo et al. synthesized a novel thermo-responsive in situ gel by functionalizing poly(N-isopropylacrylamide) with mucoadhesive gelatin and helix pomatia. One-time use of the gel in a rabbit dry eye disease model increased the drug (epigallocatechin gallate) bioavailability on the ocular surface beyond the therapeutic level for 14 days [154].

Mucolytic agents are a class of drugs that typically function by reducing mucus viscosity and inhibiting inflammatory cascades. Typically, mucolytic agents are synthetic polymers that have a thiol group, enabling them to break the disulfide bonds in mucoprotein complexes. Other types of mucolytic agents include enzymes such as papain or bromelain, which have been used as they can cleave the cross-links of mucus glycoproteins. By modifying nanocarriers with mucolytic agents, a controlled drug release, enhanced permeation, and improved mucoadhesion can be achieved. Although there has been research on the development of mucolytic enzyme-loaded nanoparticles for penetrating intestinal mucous layers, studies for ocular delivery are currently limited [155].

N-acetylcysteine (NAC) is a mucolytic agent that has been used to treat various anterior segment diseases including cataracts, DED, and filamentous keratitis. NAC is administered via intracorneal injections and is associated with side effects including edema, sloughing, and corneal haze due to the rapid mucolytic activity of NAC. Therefore, integrating them with biodegradable polymers (thiolated polymers) may improve this by achieving a controlled, sustained release. For example, the functionalization of NAC with chitosan has been an attractive strategy for forming highly mucoadhesive copolymers for ocular drug-delivery systems. By optimizing the concentration of NAC on the surface of chitosan, the thiol groups of NAC can form covalent bonds with cysteine-rich mucosal glycoproteins [156]. Nepp et al. (2020) studied the effects of chitosan-NAC eye drops (Lacrimera^TM^) for patients with dry eye disease and found a sustained improvement over a 1-month period of treatment. This was indicated by an increase in intact corneas by 64%, an increase in Schirmer’s score by 68%, and an increase in tear break-up time by 50%. Thiolated polymers also have the advantage of being able to bind with positively charged glucosamine as well as negatively charged carboxylic acids in mucosal proteins. This is because they can form amidine bonds in the case of cationic targets as well as sulfhydryl bonds in the case of anionic targets [157]. This makes thiolated polymers highly advantageous over other non-thiolated mucoadhesive polymers by significantly improving adhesion to the ocular mucus layer and therefore improving contact time with the drug. This approach was used by Sheng et al. (2022) to synthesize nanomicelles for the delivery of flurbiprofen to reduce inflammation in the context of DED. The nanomicelles demonstrated a strong binding capacity with mucin in vitro and they increased the fluidity of the ocular membranes which was presumed to be due to the ability of the NAC thiol groups to reduce intermolecular interactions between neighboring lipid molecules. The nanomicelles successfully increased the bioavailability of flurbiprofenin, an in vivo rabbit eye model [158]. However, a practical barrier with mucoadhesive nanocarriers is that rabbit models, which are most commonly used for in vivo studies, may not bring clinically translatable results given that rabbit eyes have superior bioadhesion and higher mucus production compared to human eyes [159].

In short, there are several novel drug-delivery systems (DDS) that have been developed to treat dry eye disease. Among these DDS, the nanoemulsions encapsulating cyclosporine A (Restasis^TM^) and its nanomicelle form (Cequa^TM^) are the two most used topical medications by ophthalmologists in North America. Overall, these novel DDS offer promising results in increasing drug bioavailability and reducing the need for frequent administration. This translates clinically into improved symptom management, greater compliance, and fewer side effects.

#### 5.1.2. DDS for Meibomian Gland Dysfunction (MGD)

Meibomian gland dysfunction (MGD) is a condition characterized by decreased secretion or blockage of the meibomian glands. This disruption of meibomian gland function can negatively impact both the quality and quantity of the meibum secreted, leading to changes in tear film composition, particularly the lipid layer of the tear film, and subsequent issues such as increased tear evaporation, hyperosmolarity, inflammation, and damage to the ocular surface. Currently, the main treatment strategies for MGD involve a combination of heat therapy, massage, and lid margin hygiene. Artificial tears and topical steroids can provide relief for symptoms of dry eye and ocular irritation, but their effectiveness may be limited by low patient compliance.

Chronic inflammation and oxidative stress play a critical role in the development and progression of MGD [160]. Thus, preservative-free fluorometholone eyedrops can be considered for the administration of drugs to treat MGD. However, multiple installations would be required, and they have the potential for adverse effects if used without DDS [161]. Therefore, Choi et al. developed polyhydroxyethyl methacrylate-based contact lenses embedded with cerium oxide nanoparticles for the scavenging of reactive oxygen species. The contact lenses enhanced the viabilities of human conjunctival epithelial cells and human meibomian gland epithelial cells even in media with high H_2_O_2_ concentrations. Moreover, in vivo experiments demonstrated that contact lenses had protective effects in a mouse model when 3% H_2_O_2_ eyedrops were administered [162].

Recently, nanoemulsions encapsulating cyclosporine A (nano-cyclosporine; Cyporin N^TM^, Taejoon, Korea) have been investigated in Phase 3 clinical trials for the treatment of MGD. Nanoemulsions maintain optical transparency and are considered more thermodynamically stable compared to normal emulsions. The trial concludes that cyclosporine nanoemulsions show significant improvement in dry eye disease progression secondary to MGD compared to the control group. The group receiving cyclosporine nanoemulsions had better corneal staining and increased lipid layer thickness after one month of treatment compared to the group receiving the conventional cyclosporine formulation [163].

### 5.2. DDS for Conjunctivitis

Conjunctivitis is an inflammation of the conjunctiva, the clear outer membrane that covers the sclera of the eye and the inner surface of the eyelids. This condition can be caused by viral infections, bacterial infections, allergens, irritants, or a combination of these factors. Treatment for conjunctivitis depends on the underlying cause and severity of the condition. Options can include, but are not limited to, artificial tears, topical antibiotics, corticosteroids, and immunosuppressants.

#### 5.2.1. Clinical Studies

Cyclosporine A nanoemulsions are currently being investigated in phase III clinical trials. These nanocarriers, being cationic emulsions, have the advantage of interacting with the negatively charged ocular surfaces, resulting in increased residence time. The formulation showed improved signs and symptoms of severe vernal keratoconjunctivitis, as well as good biocompatibility, except for instillation site pain [164].

#### 5.2.2. Preclinical Studies

Tacrolimus is another topical immunosuppressant clinically used to treat ocular inflammatory conditions, including vernal keratoconjunctivitis. A nano-based biodegradable DDS that can increase the stability of tacrolimus in aqueous solutions has been developed from solid lipid nanoparticles. These nanoscale particles made of solid lipids (natural fats or oils) are advantageous for encapsulating lipophilic molecules inside the lipid matrix, which improves drug solubility. The solid lipid nanoparticle in situ gels demonstrated thermo-responsive gelation at 32 degrees and had advantageous therapeutic effects in vivo compared to conventional conjunctivitis eye drops [165].

To prolong the release of topical antibiotics in the treatment of bacterial conjunctivitis, several novel DDS are being investigated in vitro and in vivo. Chitosan and PVA nanofibers have been designed to encapsulate ofloxacin. The linking of the nanofibers by glutaraldehyde vapor could reduce the burst release of ofloxacin and significantly increase its bioavailability compared to the currently available suspensions [166]. Another important strategy is to co-deliver multiple guest compounds to better target the underlying disease mechanisms. Deepthi et al. invented a copolymeric hydrogel from chitosan and poloxamer 407 for the co-delivery of neomycin (an antibiotic) and betamethasone (an anti-inflammatory compound). The hydrogels increased the bioavailability of the drug guest molecules and can reduce the frequency of dosing that is currently required for conjunctivitis eye drops [167].

### 5.3. DDS for Keratoconus

Keratoconus is a progressive eye condition in which the normally round cornea becomes thin and cone-shaped. This shape change can lead to distorted vision, nearsightedness, and irregular astigmatism. Early treatment, such as rigid contact lenses or corneal crosslinking (CXL), can help slow the disease process. Corneal crosslinking (CXL) is a surgical procedure that aims to strengthen the cornea by creating new chemical bonds within the cornea’s collagen fibers. It involves performing an epithelial debridement with a blade, applying riboflavin drops to the eye, and subsequently exposing the cornea to ultraviolet light. The traditional CXL technique has a limitation in that the cornea must be of a certain thickness to avoid making the cornea even thinner during the epithelial debridement step, which can lead to postoperative corneal ectasia. Fortunately, advanced drug-delivery systems can potentially eliminate the need for mechanical epithelial debridement, reducing the risk of complications and making the procedure safer, especially in patients with thin corneal thickness. In fact, the advanced nano-sized DDS can penetrate the cornea and target the photosensitizing agent directly to the cornea’s deeper layers.

#### 5.3.1. Preclinical Studies

Nanocarriers loaded with riboflavin have been studied for their ability to penetrate the tear film and corneal epithelium and reach the corneal stroma. Nanostructured lipid carriers, consisting of a solid lipid matrix with liquid lipid, have advantages over solid lipid nanoparticles, such as improved stability and loading capacity. These nanostructured lipid carriers have been loaded with riboflavin and displayed advantageous sustained release compared to eye drops and solid lipid nanoparticle formulations [168]. Another thermo-responsive gel for the co-delivery of dexamethasone and riboflavin has been developed from poloxamer 407 and hydroxypropyl methylcellulose. The gel displayed a therapeutic role in the increasing thickness of keratoconus and corneal fibroblast cells [169].

The delivery of certain peptides might be important for modifying keratoconus disease mechanisms. However, with the lack of appropriate DDS, the rate of diffusion and residence time of peptides in the cornea are the major barriers to their therapeutic efficacy. Copolymeric nanoparticles from chitosan-tripolyphosphate and chitosan-Sulfobutylether-β-cyclodextrin have been synthesized for the delivery of lactoferrin, a peptide that can potentially promote corneal healing. The nanoparticles displayed advantageous mucoadhesive properties that allowed them to achieve an ocular retention time of more than 240 min [170].

### 5.4. DDS for Keratitis

Keratitis is an inflammation of the cornea. The causes of keratitis can range from infectious origins (e.g., bacterial, fungal, viral infections) to autoimmune etiologies (e.g., peripheral ulcerative keratitis secondary to rheumatoid arthritis). In severe cases, it can lead to corneal melting, perforation, or scarring resulting in severe vision loss, making prompt and proper treatment important. The treatment of keratitis varies according to its etiology, with options including but not limited to topical corticosteroids, antibiotics, and immunomodulatory agents.

The only polymeric DDS approved for the treatment of keratitis is a gel formulation synthesized from sodium hydroxide, mannitol, and benzalkonium chloride for the delivery of ganciclovir [171]. The gel formulation allows for the solubilization of ganciclovir better than hydrophobic emulsions and increases the drug’s contact time within the eye [172]. This allows for ganciclovir to achieve its inhibitory concentration against the herpes simplex virus in the cornea. Nevertheless, the formulation must be applied five times a day. Non-mucoadhesive nanocarriers have the capacity to increase the ocular retention and bioavailability of ganciclovir, potentially reducing its frequent administration [173].

#### 5.4.1. Preclinical Studies

Levofloxacin is a broad-spectrum antibiotic used to treat infectious keratitis. To enhance its precorneal residence time, Jain et al. designed an in situ gel from hydroxypropyl methylcellulose and sodium alginate. The hydrogels spontaneously self-assemble at corneal pH and had a higher permeation compared to marketed Quixin^TM^ eye drops with minimal in vivo toxicity [174].

Amphotericin B is a polyene antibiotic that can self-aggregate, resulting in reduced bioavailability and reduced biocompatibility. Poly(vinylpyrrolidone) and polyvinyl alcohol microneedles have recently been developed for the ocular delivery of amphotericin B. Microneedles offer an improved solution for amphotericin B ocular treatment as they do not contain deoxycholate, eliminating the painful side effects associated with current options. Moreover, the microneedles were more effective in targeting Candida species compared to the liposomal amphotericin B formulation [175]. Another important DDS has been synthesized using hydroxypropyl methylcellulose with PEG and Poly(vinylpyrrolidone) for the delivery of moxifloxacin. The in situ gel formation prolonged the adhesion of the drug to the cornea and enabled better drug permeation compared to current commercial forms [176]. Finally, Li et al. have demonstrated the advantages of carboxymethyl-alpha-cyclodextrin conjugated with chitosan to increase the biocompatibility and aqueous stability of the econazole (an antifungal medication). The polymeric matrix increased the relative ocular bioavailability by 29 times compared to the conventional eye drop controls [177].

### 5.5. Nano-Based DDS for Cataracts

Cataract surgery, the replacement of the diseased lens with a synthetic intraocular lens, remains the treatment of choice for cataracts. It is a commonly performed procedure with a high success rate. However, like all surgical procedures, it can carry certain risks and complications, such as corneal edema, cystoid macular edema, endophthalmitis, and retinal detachment. The FDA has approved the use of topical NSAIDs for the prevention of postoperative cystoid macular edema [178]. However, the use of pharmacological compounds as an alternative to cataract surgery remains under development. To combat lens opacification in cataracts, various strategies aim to enhance the bioavailability of antioxidants in the lens [179]. This is because a significant factor in the onset of cataracts is the oxidation of lenticular proteins by reactive oxygen species and free radicals.

#### 5.5.1. Preclinical Studies

The use of silver moieties in the synthesis of nanoparticles has been explored for enhancing the topical delivery of drugs, including antioxidants. This approach aims to improve the bioavailability of the antioxidants, thereby potentially offering a new avenue for the treatment of cataracts [180,181]. This is due to their large surface-area-to-volume ratio and relative ease of manufacturing. However, silver nanoparticles have been thought to contribute to increased reactive oxygen species in their target tissue [182]. Another group has developed mesoporous silica nanoparticles loaded with CeCl_3_ that can potentially reduce the reactive oxygen species around the lens. Mesoporous silica nanoparticles have several advantages in drug-delivery applications due to their highly tunable pore characteristic. However, the formulations were designed as systemic injections and the non-biodegradable nature of silica would result in the persistence of toxic metabolites in the blood [183,184]. Therefore, emerging DDS applications for the treatment of cataracts rely on biodegradable polymers that have a predictable release profile and increased biocompatibility.

A PLGA-based nanoformulation was recently prepared by Liu et al., to combine antioxidant curcumin and cerium oxide nanoparticles. The prepared nanoformulation could have effective antioxidant and anti-glycation potential to protect lens epithelial cells. Interestingly, the nanoformulations showed lower in vivo toxicity and increased cerium nanoparticle bioavailability in the rat eye compared to subcutaneous injections [185]. Another similar formulation from low molecular weight chitosan-coated mPEG-PLGA nanoparticles was developed for the delivery of another antioxidant, baicalin. The nanoparticles had an overall small size between 148 and 219 nm and resulted in increased cellular uptake compared to the solution group. Furthermore, in vivo tests demonstrate the ability of the nanoparticles to improve precorneal residence time and significantly enhance the activities of catalase, superoxide dismutase, and glutathione peroxidase, which can neutralize the reactive oxygen species [186].

Recently, chitosan conjugated with NAC has been used as a biodegradable nanocarrier for the delivery of drugs to the anterior segment. Lan et al. have developed nanoparticles that incorporate chitosan-NAC with hydroxypropyl β-CD. The inclusion complexes of β-CD were used to encapsulate and deliver quercetin, which has been used in the treatment of cataracts. The nanoparticles enhanced the permeability of quercetin and allowed for its delivery deeper into the corneal epithelium [187].

The use of biodegradable gels is also an attractive DDS for cataracts due to their prolonged contact with the target membrane allowing for the build-up of significantly higher amounts of permeated drug at the site of administration while maintaining the drug in its bioactive form. Bodoki et al. have used biodegradable nanoparticles composed of zein and PLGA to deliver the antioxidant Lutein to prevent the progression of cataract disease. In vivo experiments showed a significant reduction in cataract severity in rats topically treated with lutein-loaded NPs compared to the positive control [188].

Table 4 presents a summary of the studies referenced in this section.

## 6. Conclusions

In conclusion, the development of polymeric nano-based drug-delivery systems (DDS) offers promising solutions to the challenges posed by the anatomical barriers of the eye in the treatment of anterior segment diseases and glaucoma. Our literature review of the latest published studies highlights the advances in polymer science and the key findings that have contributed to the progress of these systems.

The use of biopolymers in the design of DDS offers the potential to enhance therapeutic options and better manage patients with anterior and glaucomatous diseases. The high bioavailability and longer residence time of these polymeric nanocarriers in ocular tissues, combined with their biodegradable nature, make them an attractive alternative to conventional treatments for ocular pathologies.

This review highlights the current state of the field and provides insights into the future direction of research in this area. As the field continues to advance, we can expect to see further therapeutic innovations in polymeric nano-based DDS and a greater understanding of the material science aspects that are crucial for their design and development. The potential benefits of polymeric nano-based DDS in improving patient outcomes make them an exciting area of ongoing research and development.

## Figures and Tables

**Figure 1 polymers-15-01373-f001:**
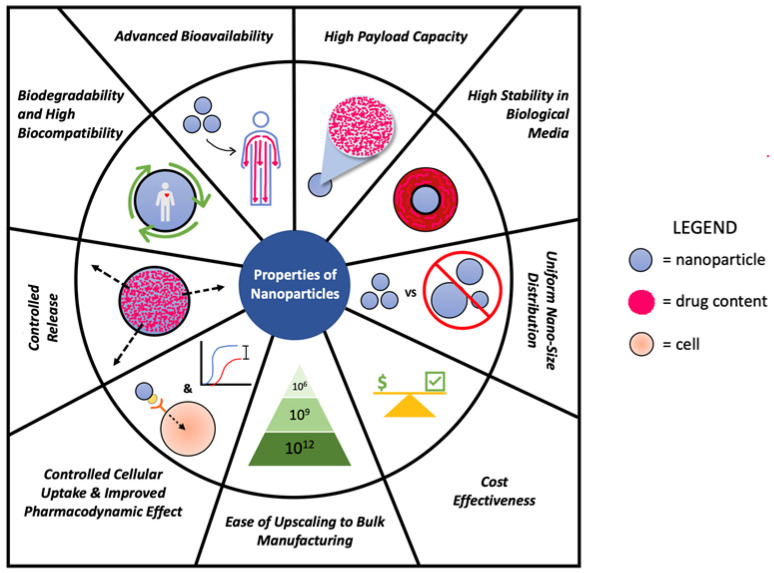
Ideal characteristics of nanocarriers.

**Figure 2 polymers-15-01373-f002:**
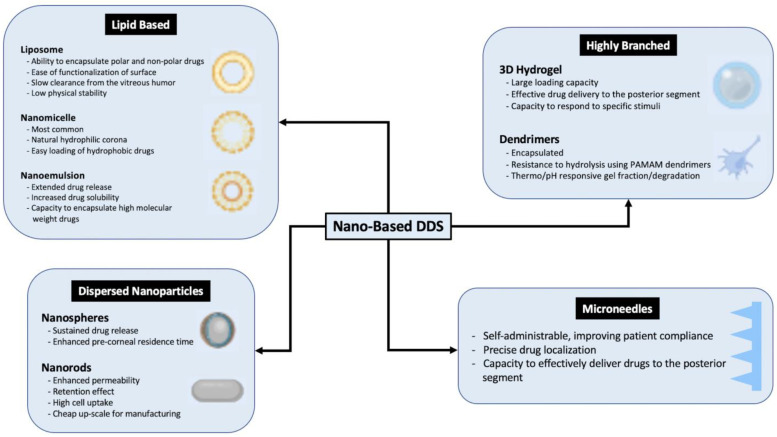
A comparative overview of characteristics and advantages of various drug-delivery systems for ocular drug delivery.

**Figure 3 polymers-15-01373-f003:**
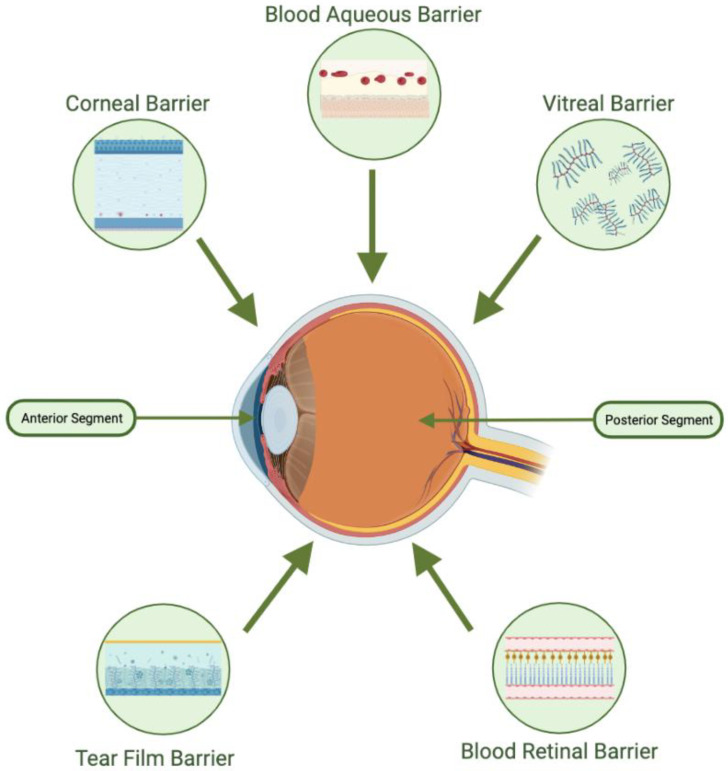
Anatomical and physiological factors affecting ocular drug delivery.

**Table 1 polymers-15-01373-t001:** Advanced DDS for IOP lowering.

DDS Technology (Polymer)	Drug	Advantages and Considerations	Administration Route	Stage	Reference	
**Niosome-based polymers**						
Chitosan-coated Niosomes	Carteolol	-Increased retention time, gradual and sustained regulation of IOP -Relative Low-cost production -Biodegradable, chemically stable -Limited shelf life	Topical	Preclinical- ex vivo and in vitro, in vivo		[56,57,58]
Niosome gel	Pilocarpine Hydrochloride					
	Latanoprost					
Proniosome gel	Brimonidine tartrate	-Physically more stable than niosomes -Improved bioavailability and prolonged release		Preclinical: in vitro and in vivo		[59,60]
	Dorzalamide-HCl					
**Nanoparticles**						
PLGA Nanoparticles	Brinzolamide	-Sustained IOP reduction -Systemic absorption observed -Minimal toxicity	Topical/ Subconjunctival	Preclinical- in vitro and in vivo		[46,47,106]
γ-cyclodextrin nanoparticle	Candesartan and irbesartan	-Polymer improves bioavailability of drug -Angiotensin receptor blockers (ARB) potentially lower IOP-IOP varied individually in study	Topical	Preclinical- in vivo		[49]
PLGA nanoparticles	SA-2 (Nitric oxide)	-Additional neuro/cytoprotective effects through increasing antioxidant (SOD) activity-Slow, sustained release		Preclinical: in vitro and in vivo		[48]
Chitosan Nanoparticles	Nanobrimonidine	-Improved synthesis of <100 nm through novel method-Improved bioavailability through mucoadhesion		Preclinical: in vitro and in vivo		[50]
CMLLG and AMLLG-based Nanoparticles	Dorzolamide	-Used natural polymer (Galactomannans) for DDS, safer and economic-Prolonged IOP reduction		Preclinical: in vitro and in vivo		[51]
PDA/PEI Nanoparticles	miR-21-5p	-Improved transfection and stability of genetic material-Low cytotoxicity with PDA addition	Intracameral injection	Preclinical- in vitro and in vivo		[52]
**Mesoporous Silica Nanoparticles**						
Hollow mesoporous organosilica (HOS) nanocapsule	Nitric oxide	-Improved biodegradability of Mesoporous Silica Nanoparticles -Prolonged NO Donors increase IOP mediated by antioxidants -Improved cornea penetration -Enhanced bioavailability	Topical	Preclinical- in vitro and in vivo		[53,54,107]
Mesoporous silica nanoparticles						
**Polymer inserts**						
Chitosan and Chondroitin sulfate polymer insert	Benzamidine (4AD)	-Novel anti-glaucomatous agent -Additional neuroprotective effect -Extended-release capacity, lowers IOP	Topical	Preclinical- in vitro and in vivo		[65,66,68]
Chitosan/hydroxyethyl cellulose insert	Dorzolamide	-Modified biodegradable polymers to carry hydrophilic drugs-Dorzolamide had additional neuroprotective effect with sustained release				
Sodium alginate+ ethyl cellulose polymer inserts	Timolol maleate					
Chitosan ocular insert	Bimatoprost	-Similar effectiveness to Bimatoprost eyedrops up to 3 weeks -Small sample size		Phase 2 Controlled study		[98]
**Polymer films**						
Chitosan film	Brimonidine tartrate	-Eco-friendly drug synthesis overcoming poor solubility of chitosan -High cornea permeability	Topical	Preclinical: in vitro and ex vivo		[67,69]
β-cyclodextrin film with PBAE and GO layers	Brimonidine	-Time-controlled drug release for precise delivery -Preliminary study		In vitro		
**Nanoemulsions**						
Nanoemulsion	Travoprost	-Enhanced absorption, prolonged IOP reduction-Long-term safety not investigated yet, toxicity with high surfactant levels and preservatives to be considered	Topical	Preclinical: in vitro and in vivo		[63,64]
	Brinzolamide		N/A	Ex vivo		
**Implants**						
Silicone Implant contact lens (IM-R lens)	Timolol, Bimatoprost and hyaluronic acid	-Preservative-free -Prevents high initial burst release -Does not affect optical properties of lenses -Large drug loss during sterilization process	Topical	Preclinical: in vitro and in vivo		[108,109]
PCL thin-film implant	Timolol and Brimonidine	-Independently controlled co-delivery -Significant IOP reduction-Systemic absorption not measured	Intracameral implant	Preclinical: in vitro and in vivo		[71,72]
	DE-117 Hypotensive agent	-Proprietary hypotensive agent used -Long-term effective IOP lowering and biocompatibility -Bulky device can cause corneal endothelium damage and device migration				
Dexamethasone-PLGA copolymer implant	Bimatoprost	-Effective sustained release -Variability in biodegradation by 1-year mark in Phase 3 trial -Risk of corneal adverse reactions		FDA-Approved-Durysta ™		[99,100,101,110]
**Gel-based polymers**						
Hybrid: PLGA-TPGS Nanoparticles in situ P407 gel	Brimonidine tartrate	-Stabilized nanodispersion using thermosensitive gel system	Topical	Preclinical- in vitro and in vivo		[74]
Hybrid: CS-SA Nanogel	Timolol Maleate	-Novel nanocarrier method with increased stability -Sustained release, increased corneal permeation		Preclinical: in vitro, ex vivo		[75]
Hybrid: Chitosan-based hydrogel	Curcumin nanoparticles and Latanoprost	-Thermosensitive -Sustained release -Preclinical cytotoxicity studies required		Preclinical: in vitro and in vivo		[73,76,78,79,81]
P407/P188 gel	Timolol Maleate					
Chitosan-based hydrogel	Pilocarpine		Intracameral injection			
Hybrid: Nanovesicles in P407/Carbopol 934P gel	Bimatoprost	-Thermosensitive gel -Sustained release and effective in lowering IOP -No irritation, inflammation observed	Topical and Subconjunctival injection	Preclinical: ex vivo, in vitro and in vivo		[77]
**Liposomes**						
Gelatinized core liposomes	Timolol Maleate	-Improved entrapment and stability -Prolonged IOP reduction	Topical	Preclinical: in vivo and in vitro		[82,83,84]
TPGS nanoliposomes	Brinzolamide					
Liposomes	Latanoprost and Thymoquinone		Subconjunctival injection			
Nanoliposome	Dorzolamide	-Extended duration of release -Complaints immediate of irritation and redness in both treatment groups	Topical	Double-blind Randomized Controlled Trial		[105]
**Dendrimers**						
PAMAM Dendrimer	Timolol	-IOP-lowering effect -No signs of cytotoxicity and ocular irritation -Further pharmacokinetic profiling required	Topical	Preclinical: in vitro and in vivo		[85,86]
	Brimonidine tartrate					
**Micelles**						
PEG-b-PPS Micelle	Peptide targeting FLT-4/VEGFR3 receptors	-Improved receptor targeting Nanocarriers -Sustained release and improvement in corneal permeability required	Intracameral injection	Preclinical- in vitro and in vivo		[89]
mPEG-PLA Micelles	Latanoprost and Timolol	-Sustained drug release, higher bioavailability -Did not affect optical properties -Lens can become rough after drug release	Topical (Contact lens)	Preclinical: in vivo		[70]
**Other**						
PC- Self-Assembly Drug Nanostructures (SADN)	Dorzalamide-HCl	-Novel system-Enhanced corneal permeations-Sustained IOP lowering -No cytotoxicity data presented	Topical	Preclinical: in vitro and in vivo		[87]
Phase transition microemulsions (PMEs)	Brimonidine tartrate (BT)	-Novel system -Systemic absorption and related side effects may be prevented with method -Prolonged release	Topical	Preclinical- in vitro and in vivo		[88]
Microneedle	Hyaluronic Acid Hydrogel	-Extended IOP reduction without drugs or surgery -No significant complications -Repeated injections may cause fibrosis -Mechanistic studies further required	Suprachoroidal injection	Preclinical: in vivo		[91]
PG-HA Nanosuspensions	Acetazolamide	-Sustained drug release -Increased drug solubility -Dispersion characteristics maintained for 6 months	Topical	Preclinical: in vitro and in vivo		[90]
Ocular ring	Bimatoprost	-Effective sustained release -Reduction of IOP is lower compared to daily timolol solution	Topical	Open-label extension to Phase 2 Clinical trial		[104]
**Cubosomes**						
GMO/P407 Cubosome	Acetazolamide	-Increased corneal permeation and ocular residence time-Good biocompatibility and no signs of cytotoxicity	Topical	Preclinical- ex vivo and in vivo		[61,62]
	Timolol Maleate					
**Montmorillonite-embedded polymers (Mt)**						
Mt-Eudragit Microsphere (MIDFDS)	Betaxolol hydrochloride (BH)	-Mt-Drug complex formed through ion exchange allows for longer controlled release of drug -DDS decreases cytotoxicity and hemolysis of drug	Topical	Preclinical- in vitro and in vivo		[92,93,94,95,111]
Mt-Solid Lipid Nanoparticles	Betaxolol hydrochloride (BH)					
Mt-PVA hybrid polymer	Brimonidine					
Mt/Chitosan Nanoparticles	Betaxolol Hydrochloride					
**Electrospun polymers**						
PVA-Poloxamer 407 Nanofiber films	Timolol maleate	-Increased retention, drug loading, and sustained IOP lowering -All biocompatible sterilization for implants needs further investigation	Topical	Preclinical- In vitro and in vivo		[96,97,112]
Lutrol + PCL ocular implants	Acetazolamide					
SA-PVA nanofibers	Forskolin					

**Table 2 polymers-15-01373-t002:** Advanced DDS for neuroprotection.

DDS	Drug	Advantages and Considerations	Administration Route	Stage	Reference
**Nanoparticles**					
PEGylated Nanoparticles	Memantine	-Reduced RGC loss -Twice daily administration -Improved drug tolerability	Topical	Preclinical: ex vivo, in vitro, in vivo	[127]
Gemini (PGL) Nanoparticles	Peptide-modified Gemini surfactants	-Non-invasive gene delivery system -Inhibited Aβ40 aggregation -Further studies on ocular in vivo models required	Topical/ Intravitreal injection	Preclinical: in vitro and in-silico	[128,129]
Polydopamine nanoparticles	Brimonidine	-Removed ROS, Promoted RGC survival, suppressed retinal inflammation -Enhanced permeability and retention	Intravitreal injection	Preclinical: in vitro and in vivo	[114]
Nanoparticles	Oncostatin M and Ciliary Neurotrophic Factor	-Sustained delivery -RGC protection -NP aggregation around injection site			[115]
PEG-PSA Nanoparticles	Brinzolamide and miRNA-124	-Prevent RGC damage and IOP-lowering effect -Sustained release, non-toxic -Mechanistic studies further required			[131]
Magnetic Nanoparticles	Neurotrophic factors (NGF, BDNF)	-Prevented RGC loss with lower dosage -Sustained, targeted release -Zebrafish model, difficulty in clinical translation			[117]
Micelle (PAMAM–PVL–PEG) Nanoparticles	DHEA and S1R agonist	-Novel nanoplatform -Effective RGC protection with targeted delivery -Needs further development for more efficient release and entrapment			[132]
Chitosan-Hyaluronic acid CS/HA nanoparticles	Epoetin Beta (EPOβ)	-Increased bioavailability -No local or systemic adverse side effects -Further therapeutic efficacy for neuroprotection in glaucoma model required	Subconjunctival injection	Preclinical: in vivo	[133]
**Microspheres**					
PLGA Microspheres	Dexamethasone and Melatonin/ CoenzymeQ10	-Reduced RGC loss -Multitherapy reduced retinal stress of single doses -Good tolerance	Intravitreal injection	Preclinical: in vitro, in vivo	[118,119,120]
	GDNF/Vit E	-Sustained controlled release for up to 6 months -Effective neuroprotection -Optimal dosage not quantified			
**Gel-based polymers**					
Benzoic-acid derivative-modified Chitosan thermogel	Pilocarpine/RGFP966	-Sustained, controlled drug release -Antioxidant, anti-inflammation properties prevent neurodegeneration	Intracameral injection	Preclinical: in vitro and in vivo	[122,123]
Dendrimer in thermogel	Pilocarpine/Ascorbic acid				
Nanoparticle-laden hydrogel	CBGA (Cannabigerolic acid)	-Improves bioavailability of drug -Reduced irritancy, improved permeation -Therapeutic efficacy with in vivo models required	Topical	Preclinical: in vitro	[124]
**Other**					
Solid Lipid Nanoparticles	Δ9-Tetrahydrocannabinol-valine-hemisuccinate	-Neuroprotective and also lowers IOP -Prolonged residence time -Further cytotoxicity profiling required	Topical	Preclinical: in vitro and in vivo	[125,126]
Microcrystals	Sunitinib	-Prevents RGC death -Therapeutically relevant concentrations obtained -Sustained release	Subconjunctival injection	Preclinical: in vivo	[134]
Cubosomes	LM22A-4 (Neurotrophic factor)	-Prevented RGC loss and improved functional outcomes -Gradual targeted release	Intravitreal injection	Preclinical: in vitro and in vivo	[121]
LAPONITE synthetic clay	Brimonidine	-Sustained delivery (up to 6 months) -Delayed neuroprotection and IOP lowering -Brimonidine depressant side effects on CNS, pharmacodynamic adjustment required	Intravitreal injection	Preclinical in vivo	[116]

**Table 3 polymers-15-01373-t003:** Advanced DDS for glaucoma surgery postoperative care.

DDS	Drug	Advantages and Considerations	Administration Route	Stage	Reference
**Polymer implants**					
Ocular implant	Bimatoprost	-Effective in lowering IOP post-SLT -Sustained release -Small sample size	Intracameral injection	Preclinical- in vivo	[136]
Collagen matrix implant	Bevacizumab and sodium hyaluronate	-Anti-scarring and wound healing post-SLT -Did not improve tissue repair clinically -DDS release requires further research	Intrableb administration	Preclinical- in vivo	[137]
**Gel-based polymers**					
Chitosan and Hyaluronic Acid-based hydrogel	5-fluorouracil (5-FU) and mitomycin C (MMC)	-Anti-scarring post-SLT -MMC had prolonged release compared to 5-FU -Cytotoxic study further required	Topical	Preclinical- in vitro	[138,139]
HECTS-AZ Hydrogel	Heparin				
Gelatin-based hydrogel	siRNA (siSparc)	-Reduced subconjunctival scarring post-SLT -Non-cytotoxic -Dose optimizing required	Subconjunctival injection	Preclinical- in vitro and in vivo	[140]
**Nanoparticles**					
DexNP γ-cyclodextrin Nanoparticles	Dexamethsaone	-Anti-inflammatory and anti-fibrotic post-SLT -Non-inferior to standard MMC treatment -Small sample size	Topical	Randomized clinical trial (single-masked)	[142]
LbL Nanoparticles	siRNA (siSparc)	-Anti-fibrotic post-SLT -Targeted delivery -Low toxicity	Subconjunctival injection	Preclinical- in vivo	[141]
**Other**					
PLGA film	5-fluorouracil and mitomycin C	-Anti-fibrotic post-SLT -Sustained effective release with lower doses -Long-term efficacy studies required	Subconjunctival route	Preclinical- in vivo	[143]

**Table 4 polymers-15-01373-t004:** Advanced DDS for anterior segment diseases.

Disease	Drug	DDS	Administration Route	Advantages and Considerations	Stage (Currently Used vs. Clinical Trial vs. Preclinical Trial)	Reference
Dry eye and corneal ulcer	Levofloxacin	HPMC and sodium alginate In situ gel	Topical	-pH-induced gel formation -Adjustable viscosity -High corneal permeability	Preclinical	[174]
Dry eye	cyclosporine A	mPEG-PLA micelles		-Extended shelf life -Reduced leakage of loaded drug -Larger drug retention in rabbit retina compared to the currently approved emulsion	Preclinical	[23]
Dry eye	Cyclosporine A	Cholesterol-hyaluronate nanomicelles	Contact lens	-Enhanced mechanical strength and wettability -Improved optical transmittance	Preclinical	[150]
Dry eye	Corticosteroid loteprednol etabonate	F127 nanomilled mucus penetrating nanoparticles	Topical	-Drug delivery surpasses the cellular inhibitory concentration of the carrier drug in vivo	Approved	[148]
Dry eye	Cyclosporine A	(HCO-40) and octoxynol 40 (OC-40). Polyvinylpyrrolidone, Sodium Phospate Monobasic Dihydrate, Sodium Phosphate Dibasic Anhydrous, and Sodium Hydroxide nanomicelles	Topical	-Higher bioavailability in vivo than free cyclosporine A -Minimal systemic leaking of Cyclosporine A	Clinical phase 3 completed	[189]
Dry eye	Rebamipide	2-hydroxypropyl-β-cyclodextrin and methylcellulose) and a gel base (Carbopol) nanoparticles	Topical applied to eyelids	-Prolonged release of Rebamipide -Sustained release exceeds the commercially available Rebamipide nanoparticles	Preclinical	[151]
Dry eye	Rebamipide	HSPC and Chol were dissolved in an appropriate amount of dichloromethane Multilamellar liposomes	Topical	-Prolonged retention time of RBM liposomes in the cornea -Clear solution increases patient compliance unlike the commercially available options	Preclinical	[152]
Dry eye	FK506	mono-functional POSS, PEG, and PPG hydrogel	Topical	-Long-acting ocular delivery system in murine animal model	Preclinical	[190]
Dry eye	flurbiprofen	N-acetylcysteine-chitosan oligosaccharide-palmitic acid nanomicelles	Topical	-Strong membrane association and prolonged precorneal retention	Preclinical	[187]
Sjogren’s dry eye	dexamethasone	PLGA and HPMC	Sub- conjunctival Implant	-Slow-release rate -High patient compliance	-Phase 3 clinical trials	[191]
Conjunctivitis	Rapamycin	poly (3- hydroxybutyrate-co-3-hydroxyvalerate) microsphere	Topical eye drops	-Trigger increased tear secretion -Reduced corneal fluorescein in vivo	Preclinical	[153]
Conjunctivitis	Ofloxacin	Chitosan and PVA nanofibers	Inserts	-Sustained release pattern for up to 96 h -Reduced burst release due to optimized crosslinking -9–20-fold higher bioavailability compared to the Ofloxacin solution	preclinical	[166]
Conjunctivitis	neomycin sulfate and betamethasone sodium phosphate	poloxamer 407 and chitosan hydrogel	Topical	-High encapsulation efficiency of hydrogels -Does not cause any irritation to the blood vessels	Preclinical	[167]
Conjunctivitis	Tacrolimus	Compritol, GMS, and dichloromethane solid lipid nanoparticles	Topical eye drop	-Rigid gel formation at 32 °C -In vivo evidence of superior pharmacodynamics compared to eye drops and solid lipid nanoparticles	Preclinical	[165]
Keratoconus	Riboflavin	Stearylamine or Trancutol P nanostructured lipid carriers	Topical	-Improves corneal crosslinking -High corneal transport	Preclinical	[168]
Keratoconus	riboflavin and dexamethasone	poloxamer 407 and HPMC gel	Topical	-Increases the cell thickness in relevant in vivo model	Preclinical	[169]
Keratoconus	Lactoferrin	Chitosan/TPP and Chitosan/Sulfobutylether-β-cyclodextrin Nanoparticles	Topical	-Host-guest complexation with BCD -Corneal residence time of more than 240 min	Preclinical	[170]
Corneal ulcer	phenytoin sodium	crown ether-based nanovesicles	Topical	-Spherical nanovesicles with very high entrapment efficiency -1.78-fold increase in corneal bioavailability compared to the drug suspension	Preclinical	[192]
Infectious Keratitis	hLF 1-11	HPMC and Hyaluronic Acid mucoadhesive matrices	unspecified	-6 months of antimicrobial activity -High stability of encapsulated peptides -Entrapment efficiency can be enhanced with Trehalose	Preclinical	[193]
Infectious Keratitis	moxifloxacin	HPMC with PVP and PEG	Ocular insert	-Adjustable concentration of HPMC and PVP reduces crystallization and increases the lamination consistency -Water encapsulation leads to increased bioadhesion -In situ gel formation prolongs the adhesion to the cornea	Preclinical	[176]
Bacterial Keratitis	Ofloxacin	chitosan and PEG-coated solid nanoparticles	Topical	-Slower drug release with 63.6% of the drug released in 3 h compared to 99.55% of the drug being released from the currently used DDS -Reduced average nanoparticle size -Accelerates ocular barrier permeation and increased adherence to the epithelium.	Preclinical	[194]
Fungal Keratitis	Amphotericin B	PVP and PVA patches	Microneedle ocular patch	-Does not contain deoxycholate that renders the currently available Amphotericin B ocular treatment painful -The microneedles completely dissolve within 1 minute in the cornea -More effective in targeting Candida compared to the liposomal amphotericin B-loaded microneedle ocular patch.	Preclinical	[195]
Fungal Keratitis	Amphotericin B	PVP and HA microneedles	Ocular patches	-Overcome the reduced loading capacity, reduced mechanical strength, and potentially high cost that could be associated with liposomal Amphotericin B formulations	Preclinical	[175]
Fungal Keratitis	Econazole	Carboxymethyl-alpha-cyclodextrin conjugated with chitosan	Topical eye drop	-Increased ocular bioavailability in the cornea by 29-times compared to controls in vivo and ex vivo after a single administration	Preclinical	[177]
Fungal Keratitis	Natamycin	Precirol ATO 5 and Pluronic f68 solid nanoparticles	Topical	-Extended drug release profile of 10 h -Increased corneal permeation compared to currently used formulation	preclinical	[196]
Keratitis and posterior uveitis	tacrolimus	PLGA, Tween, Cremophor, E80 solution, and PVA Nanocapsules	Topical eye drops	-Highly stable lyophilized DDS. -High retention and permeation of drug -Reduction in several inflammatory markers in the anterior chamber	Preclinical	[196]
Meibomian gland dysfunction	Cerium oxide nanoparticles	polyhydroxyethyl methacrylate contact lenses	Contact lenses	-In vitro and in vivo reduction of oxidative stress -Prolonged release	Preclinical	[162]
Meibomian gland dysfunction	Cyclosporin A	(Nano-cyclosporine; Cyporin N, Taejoon, Korea)	Topical	-Better corneal staining and increased lipid layer thickness compared to the group receiving the conventional cyclosporine formulation	Phase 3	[163]
Cataract disease	Cerium nanoparticles	PLGA-based nanoformulation	Topical	-Good biocompatibility -Avoids the need for subconjunctival injection	Preclincal	[185]
Cataract disease	Baicalin	mPEG-PLGA nanoparticles	Topical	-Improve precorneal residence time -Enhance the reduction of reactive oxygen species in vivo	Preclincal	[186]
Cataract disease	Lutein	Zein and PLGA	Topical	-Reduce cataracts in rat model -Fully biodegradable system emerging in the treatment of cataracts	Preclincal	[188]
Cataract disease	quercetin	Chitosan-N-acetylcysteine with hydroxypropyl β-CD	Topical	-Enhanced corneal permeability -Strong membrane association	Preclincal	[178]

## Data Availability

Not applicable.

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
