# Peer review of "Breaking Barriers in Eye Treatment: Polymeric Nano-Based Drug-Delivery System for Anterior Segment Diseases and Glaucoma"

_polymers, 2023, doi:10.3390/polym15061373_

Round 1

Reviewer 1 Report

Comments:

1.       Authors representing only polymeric nano-based drug delivery systems for Anterior Segment Diseases and Glaucoma. However similar studies are already available related to the current proposed title. Authors should emphasize “how this review differs from the available reports”.

2.       Section 2.2 Ideal characteristics of nanocarriers should be provided in graphical representation.

3.       Authors should discuss the mucolytic agents.

4.       Provide the chemical structures of important mucolytic agents and chemical structures of mucoadhesive polymers.

5.       Thiolated polymers show good mucoadhesive properties rather to other polymers. So, the authors must discuss the “role of thiolated polymers vs mucoadhesive” for Anterior Segment Diseases and Glaucoma.

6.        Provide the surface binding properties of polymers with anterior eye segments.

Author Response

Dear Reviewer,

We would like to express our sincerest gratitude for taking the time to review our manuscript. Your insightful feedback and constructive criticism have been invaluable in improving the quality of our work. We assure you that we have made every effort to address all of your comments to the best of our abilities.

If there are any areas where our response may be lacking or if there are any further concerns, please do not hesitate to bring them to our attention. We remain committed to producing high-quality article, and we appreciate the opportunity to receive constructive feedback from experts like you.

Furthermore, we would like to inform you that we have proofread our manuscript again, from the beginning to the end, to address any remaining errors or mistakes, particularly in regards to the English language. We hope that this will improve the readability and clarity of our work.

Reviewer’s comment #1: Authors representing only polymeric nano-based drug delivery systems for Anterior Segment Diseases and Glaucoma. However similar studies are already available related to the current proposed title. Authors should emphasize “how this review differs from the available reports”.

We appreciate your comment and would like to address your concern about the uniqueness of our review compared to other similar studies.

We would like to emphasize that our review differs from existing reports in several ways. Firstly, we have used figures, flowcharts, and tables to present the information in a more concise and comprehensive manner, making it easier for readers to understand the content. Secondly, the majority of our references are within the recent three years (since 2020), which ensures that our review is up-to-date and includes the latest findings in the field. In contrast, the latest review articles on this topic that we have seen only contain references up to 2020.

Furthermore, our main body is divided by anatomical regions/diseases, which makes it easier for readers to find information on the specific disease they are interested in. Additionally, half of our co-authors are clinicians, which gives us a unique perspective and allows us to provide a more in-depth and practical analysis of the topic. We also believe that our brief and concise descriptions of the diseases and conventional treatments before bridging them to the novel nano-based DDS for corresponding diseases are what sets our article apart. This serves to help everyone from diverse backgrounds to have a better understanding of the disease before delving into the details of how a polymeric nanobased DDS can help in these diseases and what additional advantages they can bring compared to conventional treatments. We believe that this approach helps to broaden our audience and make our review more accessible and relevant to a wider range of readers.

Lastly, we have made a lot of effort to make our review more reader-friendly and accessible. For example, we have included comprehensive but concise tables in each section, describing all the recent pre-clinical and clinical studies. This allows readers to quickly and easily locate the studies that might be of interest to them and gain access to the relevant references. We believe that this approach not only helps to make our review more organized and informative but also saves readers' time and effort in searching for relevant information.

Thank you again for your feedback, and we hope that our response has addressed your concerns.

Reviewer’s comment #2: Section 2.2 Ideal characteristics of nanocarriers should be provided in graphical representation.

Thank you for providing this constructive feedback. We appreciate your suggestion regarding Section 2.2, and we agree that providing a graphical representation of the ideal characteristics of nanocarriers could enhance the clarity and impact of our presentation. We have added Figure #1 to Section 2.2 of our manuscript.

Reviewer’s comment #3: Authors should discuss the mucolytic agents.

We appreciate your valuable feedback and suggestion regarding the mucolytic agents in our manuscript. We completely agree with you that this was an important aspect to consider, and we incorporated a discussion on mucolytic agents in our revised manuscript.

Mucolytic agents such as N-acetylcysteine have been reported for the treatment of some ocular diseases such as filamentous keratitis. Mucolytic agents and NAC have been discussed in section 5.1.1. The most data about the advantages of mucolytic agents have been from clinical trials in DED which is why they were discussed under that section.

Reviewer’s comment #4: Provide the chemical structures of important mucolytic agents and chemical structures of mucoadhesive polymers.

Thank you for your valuable comment regarding the inclusion of chemical structures of important mucolytic agents and mucoadhesive polymers in our manuscript. We appreciate your suggestion and agree with you that this information could enhance the understanding of our readers.

We want to inform you that we have included the chemical structures of mucolytic agents and mucoadhesive polymers in our revised manuscript. Specifically, the chemical structure of mucolytic agents and the role of the thiol group were discussed in section 5.1.1. Furthermore, we discussed the chemical structure of mucoadhesive polymers, including both mucoadhesive and thiolated polymers, in section 5.5.1.

We hope that the addition of these chemical structures will help readers to better understand the mechanisms of action of these agents and polymers in ocular drug delivery.

Reviewer’s comment #5: Thiolated polymers show good mucoadhesive properties rather to other polymers. So, the authors must discuss the “role of thiolated polymers vs mucoadhesive” for Anterior Segment Diseases and Glaucoma.

Thank you for this valuable comment. We recognize the importance of discussing the role of thiolated polymers compared to other mucoadhesive polymers in the context of Anterior Segment Diseases and Glaucoma.

We would like to inform you that we have added a discussion on mucoadhesive polymers in the treatment of Dry Eye Disease (DED) and cataract in sections 5.1.1 and 5.5.1, respectively. We have also added the relevant papers to the respective tables.

Furthermore, we agree that thiolated polymers have advantages as a mucoadhesive drug delivery system. Hence, we have included a discussion on the properties and potential applications of thiolated polymers in drug delivery after the section on mucolytic agents in section 5.1.1.

Reviewer’s comment #6: Provide the surface binding properties of polymers with anterior eye segments.

We appreciate your interest in the surface binding properties of polymers with the anterior eye segments.

We would like to inform you that we have extensively discussed the surface binding properties of polymers with the anterior eye segments in our manuscript. Specifically, we discussed the cationic and anionic interactions of polymers with mucosal proteins in response to a previous comment. Additionally, we have made further additions throughout section 2.4 to address the binding properties of polymers with the anterior eye segments.

We hope that these additions have provided a better understanding of the surface binding properties of polymers with the anterior eye segments. If you have any further suggestions, we would be glad to hear them.

Reviewer 2 Report

This research reported the amphiphilic nano-micellar system for efficient glipizide delivery, Lauric acid-conjugated-F127 (LAF127) copolymer is synthesized by esterification reaction. The particle size and morphology are also characterized, importantly, the micelles of LAF127 do not show any sign of toxicity in healthy rats. The in vitro release studies of glipizide from GNM1 depicted a sustained release profile. Overall, this is very interesting research to test common LAF127 with drug release effects. However, some categories of polymer characterizations and applications are too rough, and the content is not finely summarized. I will reconsider my suggestion after you submit a new revision.

1)     Figure 3 is not clear, the author should carefully mark all the peaks and showing the difference.

2)     The authors should further explain the reasons for the formation of morphology.

3)     Why does the size of this micelle not change according to the temperature?

4)     How does the drug loading effect of this micelle compare to other polymers?

5)     Some of the advantages of this material need to be clarified, such as degradability, stability, etc. Some paper should be cited, such as Biomacromolecules 2021, 22 (2) , 732-742. Biomolecules 2022, 12(5), 636.

Author Response

We would like to extend our thanks to you for taking the time to review our manuscript. Your insightful feedback and constructive criticism have been tremendously helpful in improving the overall quality of our work. We assure you that we have made every effort to address all of your comments to the best of our abilities.

If there is anything missing or not satisfactory, please do not hesitate to bring it to our attention. We appreciate the opportunity to receive constructive feedback from experts like you. We would also like to inform you that we have proofread our manuscript once again to ensure that all English mistakes have been corrected. We hope that this will improve the readability and clarity of our work.

Please find all of our specific responses to your comments below. We are confident that the changes we have made will address any concerns you may have had, and we hope that you find our revised manuscript to be much improved.

Reviewer’s comment #1: Figure 1 is not clear, the author should carefully mark all the peaks and showing the difference.

We appreciate your feedback regarding Figure 1 and its clarity.

We would like to inform you that we have made several adjustments to Figure 1 in response to your comment. Our aim with this figure is to provide a simplified characterization of the different nano drug delivery systems used in ocular applications, and we believe that the suggested changes have helped to clarify the differences between the various systems.

Specifically, we have carefully marked all the peaks and highlighted the differences between similar drug delivery systems. For example, we have included information on nanomicelles, which have been reported as the most commonly used system, and the natural hydrophilic corona that distinguishes nanomicelles from liposomes has also been added as it is important for the drug delivery mechanism.

We hope that these adjustments have improved the clarity of the figure and made it easier for readers to understand the different drug delivery systems used in ocular applications. Once again, we appreciate your feedback, which has undoubtedly helped us to improve the quality of our work.

Reviewer’s comment #2: The authors should further explain the reasons for the formation of morphology.

Thank you for your valuable comment. We appreciate your input and agree that further explanation of the reasons for the formation of morphology could enhance the understanding of our readers. We have made additions to our revised manuscript in different sections to address this concern. The advantages of different morphologies have been mentioned in section 2.3. The advantages of nanoparticle morphology (nanorods vs nanospheres) have been explained in section 2.4.2.

The reasons for the formation of morphology have been explained for different nanoparticles as follows:

2.5.1 Nanomicelles

No changes. This was already discussed in that section.

2.5.2 Liposomes

- These vesicles are composed of one or more phospholipid bilayers that self-assemble into spherical vesicles with a hydrophilic core in aqueous media

2.5.3 Dispersed nanoparticles

- These structures self-assemble into different morphologies and this process is highly dependent on the nature of the polymers that have been chosen. It is not possible to come up with a generalized explanation for their formation due to the effect of: shape of template polymer, solvent type, surface charge of polymers, type of cross-linker used, type of polymer blending reaction chosen, and purification method. In practice, the morphology of nanoparticles after blending the polymers is often unpredictable.

2.5.4 Dendrimers

- Dendrimers are formed of radially symmetric and branched molecules that cross-link several spherical layers. Growing outwards, the core interacts with monomers containing a reactive and two dormant groups, which activates a cascade reaction to form peripheral branches. They have the capacity to encapsulate different drugs in their large central core.

2.5.6 Hydrogels

- Various chemical and physical cross-linking between copolymers allow for the formation of molecules with large water-accommodating pores. The surface polarity and solubility of the chosen polymers typically influence the morphology of the pores.

2.5.7 Microneedles

No change. These are larger systems and sometimes the morphology is based on macro-sized templates.

Reviewer’s comment #3: Why does the size of this micelle not change according to the temperature?

Thank you for your comment and question regarding the temperature stability of nanomicelle size. We appreciate your feedback and your guidance in focusing our discussion. The lack of change in size is indeed due to the formation of hexagonal micellar phases by F127 with increasing temperature, as mentioned in our revised manuscript in section 5.1.1. We have also added further explanation to clarify this point. Once again, thank you for bringing this to our attention. We appreciate your attention to detail and expertise in the field. Please let us know if any of our answers were inaccurate or if you have any further questions or concerns. We value your input and strive to provide accurate and informative responses.

Reviewer’s comment #4: How does the drug loading effect of this micelle compare to other polymers?

Thank you for your question. It has been reported that F127 has a longer hydrophobic poly(propylene oxide) chains that can provide strong hydrophobic absorption on the surface of the nanoparticles. This nanoparticle formulation was compared with other commercially available loteprednol formulation and it had better pharmacokinetic effects in vivo. This information was added to the paper. However, since the formulation is patented, there is very little information available about the drug loading effect as the in vitro drug loading profile has not been reported in the literature.

Thank you for your question. We appreciate your feedback and would like to ensure that our response is accurate. Please correct us if we have provided any inaccurate information in our previous response.

Reviewer’s comment #5: Some of the advantages of this material need to be clarified, such as degradability, stability, etc. Some paper should be cited, such as Biomacromolecules 2021, 22 (2) , 732-742. Biomolecules 2022, 12(5), 636.

Thank you for your comment and for bringing those papers to our attention. We have carefully reviewed the papers you suggested and agree that they provide valuable insights into the properties of the materials discussed in our article. We have added those papers to our reference list and have incorporated some of their findings into our discussion on the advantages of the materials.

The first paper mentioned in this comment, “Efficient Synthesis of Folate-Conjugated Hollow Polymeric Capsules for Accurate Drug Delivery to Cancer Cells”, discusses an interesting drug delivery system consisting of hollow nanoporous capsules. The article was cited under hollow capsules in section 4.1.1.

We have added the second paper to section 2 and emphasized the interactions with PEG, as they are relevant to our article. However, we would like to clarify that PGA Poly(α-L-glutamic acid) (mentioned in the second paper) has not yet been reported for ocular drug delivery, and our article does not cover anti-cancer treatments. Therefore, if you feel that this addition is not appropriate, please feel free to remove it